# Silicon photocathode functionalized with osmium complex catalyst for selective catalytic conversion of $CO_2$ to methane

Xing-Yi Li [1,2,6], Ze-Lin Zhu [3,6], Fentahun Wondu Dagnaw [1], Jie-Rong Yu [1], Zhi-Xing Wu [4], Yi-Jing Chen [1], Mu-Han Zhou[1], Tieyu Wang [2], Qing-Xiao Tong [1,5] ✉ & Jing-Xin Jian [1,2] ✉

Solar-driven $CO_2$ reduction to yield high-value chemicals presents an appealing avenue for combating climate change, yet achieving selective production of specific products remains a significant challenge. We showcase two osmium complexes, przpOs, and trzpOs, as $CO_2$ reduction catalysts for selective $CO_2$-to-methane conversion. Kinetically, the przpOs and trzpOs exhibit high $CO_2$ reduction catalytic rate constants of 0.544 and 6.41 $s^{-1}$, respectively. Under AM1.5 G irradiation, the optimal Si/$TiO_2$/trzpOs have $CH_4$ as the main product and >90% Faradaic efficiency, reaching −14.11 mA $cm^{-2}$ photocurrent density at 0.0 $V_{RHE}$. Density functional theory calculations reveal that the N atoms on the bipyrazole and triazole ligands effectively stabilize the $CO_2$-adduct intermediates, which tend to be further hydrogenated to produce $CH_4$, leading to their ultrahigh $CO_2$-to-$CH_4$ selectivity. These results are comparable to cutting-edge Si-based photocathodes for $CO_2$ reduction, revealing a vast research potential in employing molecular catalysts for the photoelectrochemical conversion of $CO_2$ to methane.

With the excessive exploitation and utilization of fossil fuels, the concentration of carbon dioxide ($CO_2$) in the atmosphere increases significantly, which indirectly causes global warming, environmental pollution, and ecological destruction[1-4]. To mitigate these problems, scientists have recreated the methods of artificial photosynthesis and carbon cycle by capturing and converting $CO_2$ into storable fuels and chemicals[5-11]. Solar-driven $CO_2$ reduction ($CO_2$R) by photocatalytic (PC) or photoelectrochemical (PEC) configurations provides an ideal solution for utilizing solar energy to produce solar fuels[12-15]. However, the $CO_2$R reaction involves the continuous transfer of multiple electrons and protons, accompanied with various intermediates and terminal products, and facing with the competition of proton reduction[16]. Among the different products obtained from $CO_2$R, the

production of methane ($CH_4$) is the most challenging route for renewable fuel, which must simultaneously manage the transport of 8 electrons and 8 protons[17-19]. To date, the scalable and selective $CO_2$-to-$CH_4$ conversion in solar-driven PEC systems has yet to be accomplished[20].

To enable a practical solar-driven $CO_2$R, two essential components of semiconductors and catalysts are required[20-22]. The former harvests sunlight to generate electrons and sets an upper limit on the solar-to-fuel conversion efficiency, while the latter determines the activity and selectivity of $CO_2$R. Silicon (Si) is a commercially available semiconductor for photocathode candidates with its advantages of element abundance, low environmental hazard, efficient sunlight absorption, high saturated photocurrent, and

[1]Department of Chemistry, Shantou University, Shantou 515063, PR China. [2]Guangdong Provincial Key Laboratory of Marine Disaster Prediction and Prevention, Shantou University, Shantou 515063, PR China. [3]Center of Super-Diamond and Advanced Films (COSDAF) and Department of Chemistry, City University of Hong Kong, Hong Kong SAR, PR China. [4]Laboratory of Organic Electronics, Department of Science and Technology (ITN), Linköping University, Norrköping, SE 60174, Sweden. [5]Key Laboratory for Preparation and Application of Ordered Structural Material of Guangdong Province, Shantou University, Shantou 515063, PR China. [6]These authors contributed equally: Xing-Yi Li, Ze-Lin Zhu. ✉e-mail: qxtong@stu.edu.cn; jxjian@stu.edu.cn

industrial applicability[23–28]. Moreover, the conduction band (CB) position of Si ideally straddles $CO_2$ reduction potentials, enabling efficient $CO_2$-to-$CH_4$ conversion[29,30]. In the past decade, gratifying progress has been made on Si-based photocathodes for $CO_2R$[31–39]. In 2016, Zetian Mi's group reported that $TiO_2$-protected $n^+p$-Si photocathode with $Au_3Cu$ nanoparticles catalyst for $CO_2R$, which has a Faradaic efficiency of 70% for carbon monoxide (CO) at $-0.18\ V_{RHE}$[33]. Recently, Erwin Reisner's work demonstrated a $TiO_2$-protected Si photocathode with Co phthalocyanine catalyst for $CO_2$-to-CO conversion, which has a $J_{ph}$ of around $-0.15\ mA\ cm^{-2}$ at $-0.53\ V_{RHE}$ and CO selectivity of $66 \pm 3\%$[38]. At present, most Si-based photocathode systems produce CO and formate ($HCOO^-$) as the predominant products, and the photocurrent is relatively small. As the molecular catalyst is a key component in the photocathodes for generating selective $C_1$ products, seeking other suitable $CO_2R$ catalysts to deposit on Si-based photocathode is highly desired.

Platinum group metals (PGMs, e.g., Ru, Rh, Pd, Os, Ir, Pt) have gained great attention as redox photocatalysts in solar-driven $CO_2R$ conversion[29,40–49]. PGM complexes could harvest photons and reach the triplet excited state via a rapid intersystem crossing process[49,50]. Then, their excitons have long lifetimes to drive catalytic reactions[51,52]. Besides that, PGM complexes have sufficient redox capacity and tunable coordination ability with carbonyl ligands, which contribute to the enhancement of its activity and selectivity for $CO_2$-to-fuels conversion. Among the PGMs complexes, [Os] complexes have rare reported as $CO_2R$ catalysts in solar-driven systems[40,53].

Here, we demonstrate two [Os] complexes, namely przpOs, and trzpOs, as $CO_2R$ catalysts for highly selective $CO_2$-to-methane conversion. Kinetically, przpOs and trzpOs exhibit catalytic rate constants ($k_{cat}$) of 0.544 and $6.41\ s^{-1}$ for $CO_2R$, respectively. DFT calculations and electrochemical spectroscopy studies show that the N atoms on the bi(1,2,4-triazole) ligand provide binding sites for $CO_2$ substrate and proton, promoting the high selectivity of $CO_2$-to-$CH_4$ conversion. Under AM1.5 G irradiation, the optimal $Si/TiO_2$/trzpOs exhibits $CH_4$ as the main product and >90% Faradaic efficiency, reaching a high photocurrent density of $-14.11\ mA\ cm^{-2}$ at $0.0\ V_{RHE}$. These results are comparable with start-of-the-art Si-based photocathodes for $CO_2R$, unveiling a broad research prospect in [Os] complex for $CO_2$-to-methane conversion.

## Results

### Synthesis and characterization of [Os] complexes

Multiple nitrogen heterocyclic structure ligands of 5,5′-bis(trifluoromethyl)−$2H,2'H$−3,3′-bipyrazole ($bpzH_2$) and 5,5′-bis(trifluoromethyl)−$2H,2'H$−3,3′-bi(1,2,4-triazole) ($btzH_2$) were prepared according to literatures[54,55]. The designed triazole ligand has additional nitrogen heteroatoms that can be used as binding sites for $CO_2$ substrate and proton to promote the proton-coupled electron transfer (PCET) process. Subsequently, these bipyrazole and triazole ligands reacted with $Os_3(CO)_{12}$ and 1,10-phenanthroline (phen) to form osmium complexes of przpOs and trzpOs (Fig. 1a). The successful preparation of przpOs and trzpOs was confirmed by hydrogen nuclear magnetic resonance (¹H-NMR) spectroscopy (see Supplementary Fig. S1, S2), mass spectroscopy (MS) (see Supplementary Fig. S3) and Fourier transform infrared (FTIR) spectroscopy (see Supplementary Fig. S4). The ¹H NMR spectroscopic analysis confirmed the presence of 32 protons in complexes przpOs and 30 protons in trzpOs, each associated with their respective ligands. Specifically, the ¹H NMR spectrum of przpOs displayed a distinctive singlet at 6.76 ppm, attributed to the two protons on the 3,3′-bipyrazole ligand, while the methyl protons corresponding to the $PhPMe_2$ ligands were observed at 0.76 (s, 12H). High-resolution mass spectrometry (HR-MS) studies revealed a molecular mass of 916.1683 for przpOs, compared to that of 918.1588 observed for trzpOs. The molecular ion peak signals of both complexes were prominent in the HR-MS analysis (see Supplementary Fig. S3), with minimal ion residue peaks suggesting their stability and resistance to decomposition into ion fragments. The stretching vibration signals of aromatic C-H and methyl C-H were confirmed in FTIR spectra, and the stretching vibration characteristics of the aromatic ring skeleton were observed (see Supplementary Fig. S4). The solid-state of przpOs was further confirmed by the single-crystal X-ray structural characterization (Fig. 1b, Supplementary S1). For the side view of przpOs, two P ligands are in the axial direction with Os-P lengths of 2.3388 and 2.3429 Å, respectively, and P-Os-P dihedral angle is 174.59°. Moreover, the phenyl group in the P ligand is parallel to the phen ligand due to the π-π stacking. From the top view, the Os atom locates at the center of the plane constructed by phen and bipyrazole ligands, and the distances between Os and N atoms are 2.0852, 2.0791, 2.0760, and 2.0649 Å, respectively (Fig. 1b). It is particularly noteworthy that the uncoordinated $N_4$ atom in the structure of przpOs exposes the ideal binding sites for $CO_2$ and proton substrates.

The photophysical properties of przpOs and trzpOs were characterized by the ultraviolet-visible (UV-vis) absorption and steady-state photoluminescence (PL) spectroscopy in dichloromethane ($CH_2Cl_2$) solution. As shown in Fig. 1c, the absorption bands <350 nm mainly originated from the ligand π–π* transitions, while the absorptions ranging from 400 to 550 nm are ascribed to the metal-to-ligand charge transfer (MLCT) transitions. The board absorption bands across 550–700 nm region are attributed to their $d$–$d$ transitions[52]. Consequently, the optical band gaps ($E_{g,op}$) of przpOs and trzpOs calculated from the edge of MLCT absorption peaks are 2.30 and 2.45 eV, respectively (see Supplementary Table S1). Interestingly, trzpOs has a larger $E_{g,op}$ value than that of przpOs, indicating that its excited state possesses higher energy for transferring photogenerated charge to the $CO_2$ substrate, potentially leading to more efficient $CO_2$ reduction activity. This finding aligns with the results obtained from the electrochemical studies. The PL spectra of przpOs and trzpOs show board emissions centered at 839 and 783 nm, with quantum yields (QY) of 0.2% and 2.5%, respectively (Fig. 1c, Table S1). By contrast, trzpOs shows a blue-shift emission, which has a similar trend with its absorption character. Moreover, time-resolved PL spectra of przpOs and trzpOs in $N_2$ saturated solution display average lifetimes ($\tau_{N2}$) of 1445.4 and 1604.0 ns, respectively, which significantly decreased to 866.4 and 815.8 ns, respectively, after purging the solution with $CO_2$ (Fig. 1d, Table S2). This result indicates that the excited state of [Os] complexes could deliver photogenerated electrons to $CO_2$ substrate for its reduction process. Thus, the electron transfer constant ($k_{et}$) from [Os] complex to $CO_2$ is estimated to be $4.62 \times 10^5$ of przpOs and $6.02 \times 10^5\ s^{-1}$ of trzpOs, respectively (see Supplementary Table S2). Therefore, trzpOs have a faster electron transfer rate than that of przpOs, which is conducive to efficient photocatalytic $CO_2$ reduction.

### Electrochemical activity of $CO_2$ reduction

Cyclic voltammetry (CV) of przpOs and trzpOs in acetonitrile solvent exhibit two reversible redox peaks ($E_1$ and $E_2$) at 0.64 and $-1.44$ V versus normal hydrogen electrode ($V_{NHE}$) and 0.82 and $-1.44$ V vs. NHE, respectively (Fig. 2a, Table S3). Then, the electrochemical redox bandgap ($E_{g,redox}$) was determined to be 2.08 and 2.26 eV, respectively, which has a similar trend of their optical bandgaps. Moreover, the $E_2$ values are lower than the thermodynamic equilibrium required potentials for hydrogen evolution reaction (HER) and $CO_2$-to-$CH_4$ conversion[18], enabling that [Os] complexes could be used as photocatalysts for $H_2$ and $CH_4$ generation via proton reduction and $CO_2R$. Their electrochemical behaviors were conducted in pure $N_2$ and $CO_2$ saturated solvents to investigate their HER and $CO_2R$ activities. As shown in Fig. 2b, przpOs exhibits a reduction peak current ($i_0$) of 2.7 μA

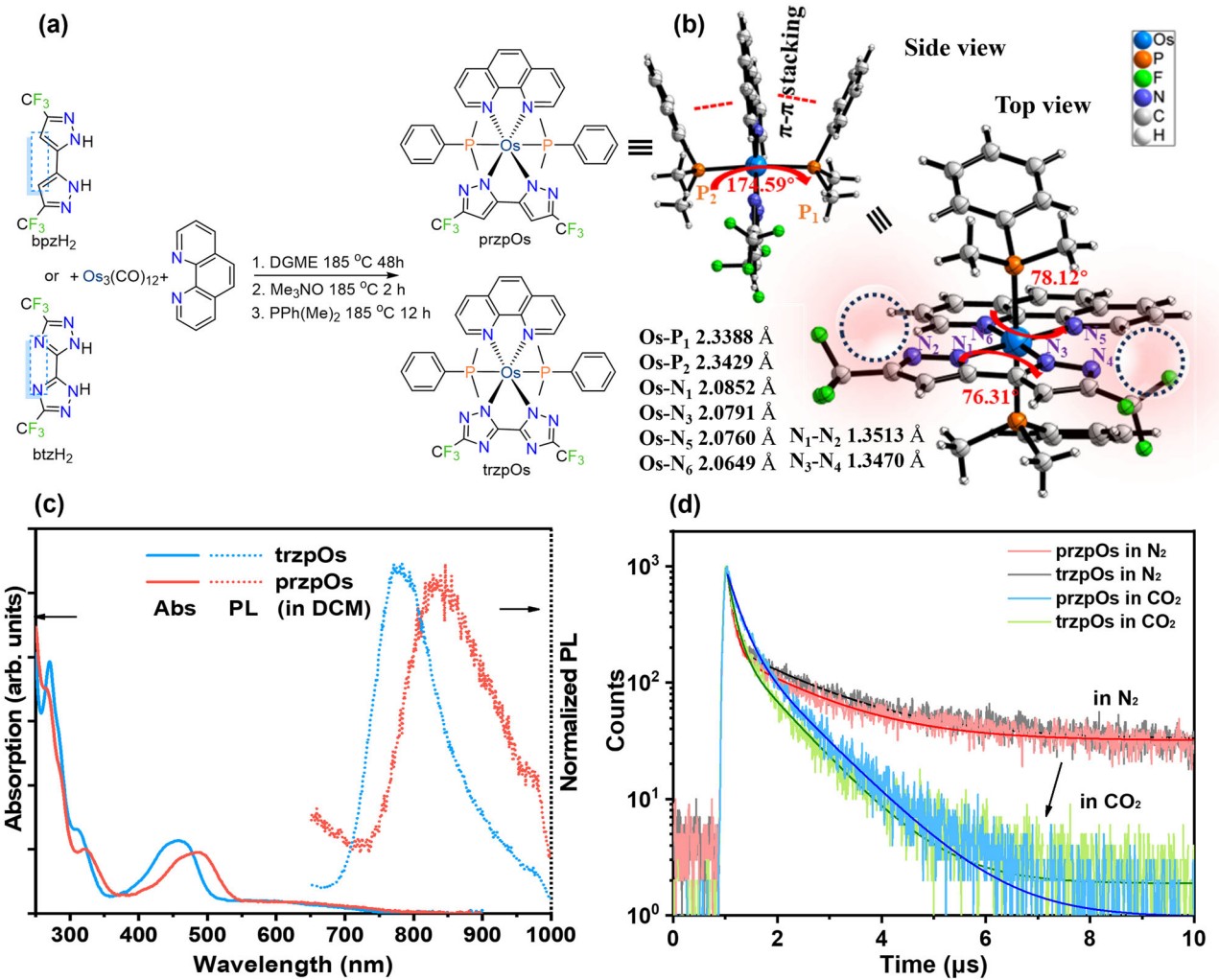

**Fig. 1 | Synthesis and characterizations of [Os] complexes. a** Synthesis of przpOs and trzpOs. **b** Single-crystal X-ray structure of przpOs from the side and top views (thermal ellipsoids: 30%). **c** UV-visible (UV-vis) absorption and photoluminescence (PL) spectra of $10^{-5}$ M przpOs and trzpOs in dichloromethane. **d** PL lifetimes of $10^{-5}$ M przpOs and trzpOs in $N_2$ and $CO_2$ atmosphere.

at $E_2$ potential in $N_2$ atmosphere. After purging $CO_2$, an irreversible reduction peak appeared at 1.3 $V_{NHE}$, which is assigned to $CO_2R$ catalytic current ($i_c$), and the corresponding current increased to 7.9 μA. In contrast, the trzpOs has a $i_0$ of 2.3 μA at $E_2$ with purging $N_2$, and $i_c$ of 7.3 μA at 1.3 $V_{NHE}$ with purging $CO_2$. In order to verify whether the catalytic process is related to protons, 1 mM trifluoroacetic acid (TFA) was added as a proton source. The catalytic current with protons ($i_{c-H}$) of przpOs and trzpOs was significantly enhanced to 18.1 and 52.9 μA, respectively. The catalytic current enhancement of trzpOs with proton source is higher than that of przpOs, indicating that the N atoms on the triazole ligand provide binding site for protons to promote the PCET process, resulting in more efficient catalytic reduction. The results also imply that $CO_2R$ is a mass transfer-controlled reaction, meaning that the reaction process could be improved by enhancing mass transferring.

Gas chromatography measurements confirmed that the main electrocatalytic $CO_2R$ product of [Os] complexes at $-1.4$ $V_{NHE}$ was $CH_4$, with trace of CO and $H_2$ (Fig. 2d–f). The $CO_2R$ activity and selectivity of trzpOs are higher than przpOs, which is consistent with their electrochemical properties. It's worth noting that both trzpOs and przpOs present repeatable photo-response currents for electrocatalysis $CO_2R$ at $-1.4$ $V_{NHE}$, indicating that [Os] complex can be used as the light harvesting units as well as the $CO_2$-to-$CH_4$ conversion catalyst. Consequently, the first-order catalytic constant ($k_{cat}$) of [Os] complex,

usually referred to the turnover frequency (TOF) of catalyst, can be assessed by Eq. (1):

$$\frac{i_c}{i_0} = \frac{n}{0.4463}\left(\frac{RTk_{cat}}{Fv}\right)^{1/2} \qquad (1)$$

where R is universal gas constant (8.314 J mol$^{-1}$ K$^{-1}$), F is Faradaic constant (96485 C mol$^{-1}$), T is the temperature (298.13 K), $v$ is the scan rate (0.1 V s$^{-1}$), and n is the number of electrons involved in the catalytic reaction (8 for $CO_2$-to-$CH_4$ conversion). Therefore, the $k_{cat}$ of przpOs is 0.104 s$^{-1}$ under neutral conditions, and $k_{cat-H}$ increases to 0.545 s$^{-1}$ under proton source conditions with TFA. In contrast, trzpOs exhibits higher $k_{cat}$ of 0.122 s$^{-1}$ without proton and $k_{cat-H}$ of 6.41 s$^{-1}$ with proton source. Electrochemical analysis has revealed distinct differences in the $k_{cat-H}$ values of przpOs and trzpOs catalysts when an additional proton source is introduced. We hypothesized that an extra nitrogen atom on the 3,3'-bi(1,2,4-triazole) ligand in trzpOs serves as a proton-relay that facilitates the proton transfer for the sequential $CO_2$ reduction process. The N atom can provide binding sites for protons, which was verified in subsequent comparative experiments. Under $N_2$ atmosphere, as TFA was incrementally added to the acetonitrile solution of przpOs, its $E_2$ peak became irreversible with a notable increase in the cathodic current, pointing towards a characteristic of the hydrogen

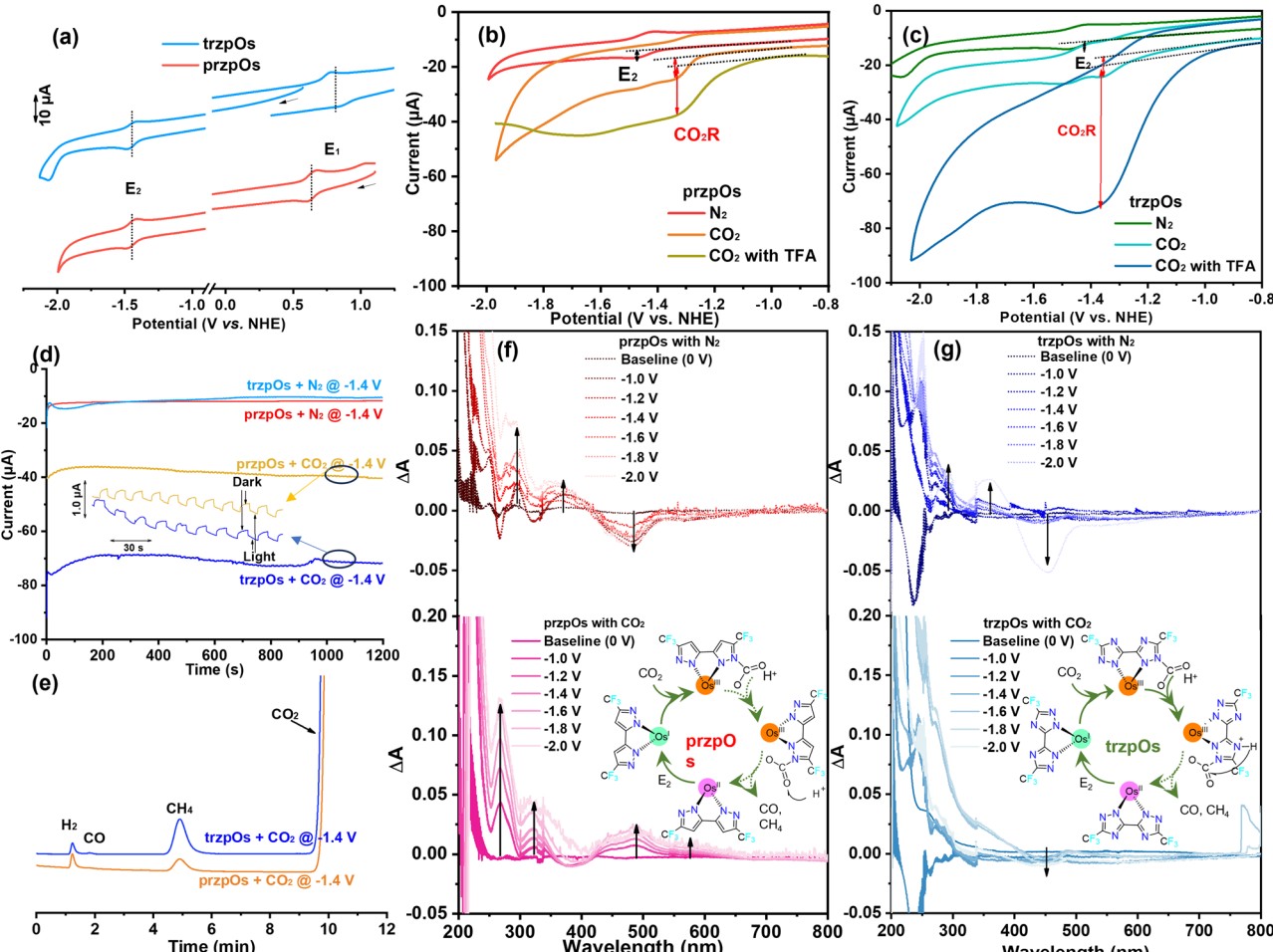

**Fig. 2 | Electrochemical behaviors and electrochemical spectra of [Os] complexes. a** CV curves of 1.0 mM of przpOs and trzpOs in electrolyte of 0.1 M n-Bu$_4$NPF$_6$ acetonitrile solution, in N$_2$ atmosphere, at scan rate of 0.1 V s$^{-1}$ and without iR correction. The arrows indicate the scanning direction. CV curves of przpOs (**b**) and trzpOs (**c**) in N$_2$ and CO$_2$ saturated electrolyte with the addition of 1.0 mM TFA. **d** Current-time curves of przpOs and trzpOs in N$_2$ and CO$_2$ at −1.4 V$_{NHE}$, under chopped AM 1.5 G illumination. **e** Gas products of przpOs and trzpOs in CO$_2$ at −1.4 V$_{NHE}$ determined by GC measurements. In situ UV-vis spectro-electrochemistry of 1.0 mM przpOs (**f**) and trzpOs (**g**) in N$_2$ and CO$_2$ saturated electrolyte at different applied potentials. The inset figure shows the proposed catalytic cycles.

evolution process (HER) (see Supplementary Fig. S5). Conversely, the HER current of trzpOs demonstrated a considerably greater enhancement compared to that of przpOs, thus confirming the superior proton-binding capability and showcasing trzpOs' heightened proton-reduction activity.

Potential-dependent UV−vis spectra of przpOs and trzpOs were carried out in the range of −1.0 to −2.0 V in N$_2$ and CO$_2$ saturated environment to detect the intermediates during the catalytic cycle (Fig. 2f, g). As the voltage decreases from −1.0 to −2.0 V, [Os] complexes in N$_2$ atmosphere exhibit enhanced absorption signals at 372 and 294 nm, 360 and 290 nm, and bleaching signals at 483 and 453 nm for the przpOs and trzpOs, respectively, which is attributed to the formation of reduced state [Os] complexes. Furthermore, these signals decrease sharply after purging CO$_2$ with other peaks appearing and gradually increasing, indicating the interaction between the excited state [Os] and CO$_2$ to form CO$_2$ adducts. Accordingly, we propose the catalytic cycle shown inset of Fig. 2f, g, the CO$_2$ molecule firstly interacts with the N atoms on the open site of bipyrazole and triazole ligands, and further forms CO$_2$ adduct via hydrogenation reaction. The CO$_2$R products (e.g., CO, CH$_4$) then desorb from active sites and diffuse to bulk solution. Compared with przpOs, trzpOs has a synergistic catalytic effect, which can promote the proton transfer from the solvent to the active sites via N atoms on triazole ligand, resulting in enhanced CO$_2$R activity.

## Photoelectrochemical CO$_2$ reduction on Si-based electrode

Subsequently, [Os] complexes were deposited on a p-n heterojunction Si/TiO$_2$ photocathode to construct solar-driven CO$_2$ conversion system. Si-based photocathode has advantages of element abundance, sunlight absorption, and high saturated photocurrent, as well as that its CB ideally straddles the required potential for CO$_2$-to-CH$_4$ conversion[30,56]. Herein, a black Si photocathode with a nano-porous surface was prepared by a PEC HF etching method developed in our previous works[57,58], which provides a large specific surface area for the deposition of abundant catalysts. Moreover, n-type TiO$_2$ layer was coated on the black Si surface by magnetron sputtering to form p-n heterojunction and protective layer, which can further improve the charge separation and stability of Si-based electrode[33,38]. The Si/TiO$_2$/[Os] photocathodes were prepared by dropping the acetonitrile solution of the [Os] complexes onto the surface of Si/TiO$_2$ electrode and drying at room temperature. The uniform distribution of Os and F elements on the surface of Si/TiO$_2$/[Os] photocathode was confirmed by the scanning electron microscopy (SEM) and the corresponding energy dispersive X-ray spectroscopy (EDX) (see Supplementary Fig. S6). In the Si/TiO$_2$/przpOs and Si/TiO$_2$/trzpOs electrode, the composition of Os element is similar, 0.95 % and 1.01 %, respectively.

Under the illumination of simulated sunlight (AM1.5 G, 100 mW/cm$^2$), Si/TiO$_2$/trzpOs photocathode exhibits enhanced photocurrents for CO$_2$R with a gradually increasing of trzpOs catalyst (see

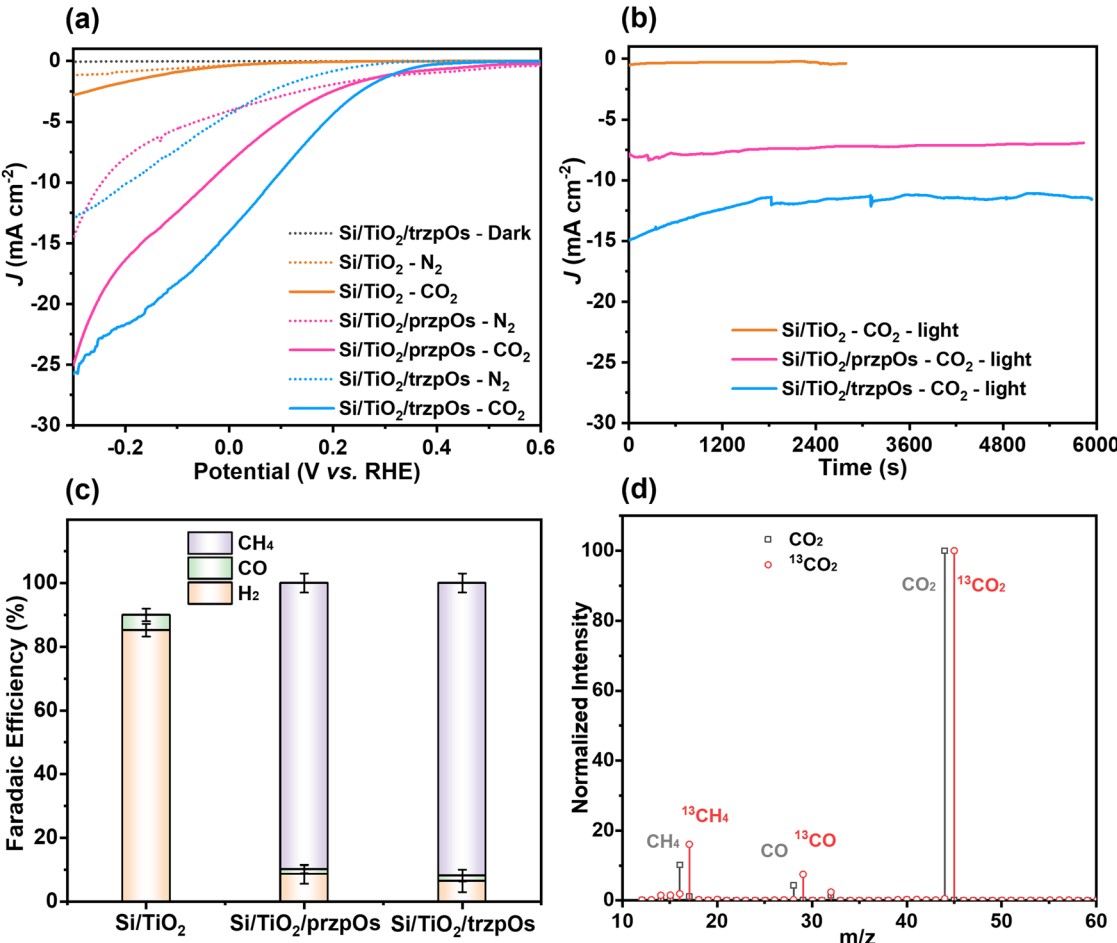

**Fig. 3 | PEC performance of [Os] complexes on Si photocathode. a** LSV curves performed by Si/TiO$_2$, Si/TiO$_2$/przpOs and Si/TiO$_2$/trzpOs electrode in N$_2$ and CO$_2$ atmosphere, in 0.5 M Na$_2$SO$_4$ (pH 6.8 ± 0.3) solution, at scan rate of 30 mV/s, with 1.6 nmol/cm$^2$ of [Os] complexes and without iR correction. *J-t* curves (**b**) and Faradaic efficiency (**c**) of Si/TiO$_2$, Si/TiO$_2$/przpOs and Si/TiO$_2$/trzpOs at 0 V$_{RHE}$. The Faraday efficiency was tested three times repeatedly. **d** GC-MS analysis of Si/TiO$_2$/trzpOs in 0.5 M Na$_2$SO$_4$ (pH 6.8 ± 0.3) solution with $^{13}$CO$_2$ and CO$_2$, respectively.

Supplementary Fig. S7). The optimal Si/TiO$_2$/trzpOs with 1.6 nmol/cm$^2$ of trzpOs catalyst exhibits a high $j_{ph}$ of −14.11 mA/cm$^2$ V$_{RHE}$ without external bias (0.0 V$_{RHE}$) and −25.8 mA/cm$^2$ at potential of −0.3 V$_{RHE}$, which are 140 and 92 times that of Si/TiO$_2$ with $j_{ph}$ of −0.02 mA/cm$^2$ at 0.0 V$_{RHE}$ and −0.28 mA/cm$^2$ at −0.30 V$_{RHE}$, respectively (Fig. 3a, Table S4). Moreover, the onset potential ($E_{on}$) for photo-response current is positive-shifted from 0.24 V$_{RHE}$ of Si/TiO$_2$ to 0.52 V$_{RHE}$ of Si/TiO$_2$/trzpOs. In contrast, the Si/TiO$_2$/przpOs photocathode shows a $j_{ph}$ −4.11 mA/cm$^2$ in N$_2$ and −8.43 mA/cm$^2$ in CO$_2$ without external bias (Fig. 3a), which is lower than values performed by Si/TiO$_2$/trzpOs electrode at the same conditions. As a systematic comparison, the Si/TiO$_2$ without [Os] complex displays low photocurrents of −0.34 and −0.39 mA/cm$^2$ at 0.0 V$_{RHE}$ in N$_2$ and CO$_2$ atmosphere, respectively (Fig. 3a). These results indicate that [Os] complexes are efficient catalysts for CO$_2$R, manifesting with the increased photocurrent and mitigated voltage loss[38].

Current-time (j-t) measurements were carried out to evaluate the CO$_2$R stability of Si/TiO$_2$/[Os] photocathodes under continuous illumination of AM 1.5 G. As shown in Fig. 3b, the photocurrent densities of Si/TiO$_2$/[Os] photocathodes stabilize within 6000 s. Meanwhile, gas products were monitored by gas chromatography (GC) analysis, presenting H$_2$, CO, and CH$_4$. Accordingly, the Si/TiO$_2$ photocathode without [Os] complex exhibits a high Faradaic Efficiency of H$_2$ (FE$_{H2}$) over 95% and a low Faradaic Efficiency of CO product (FE$_{CO}$) (Fig. 3c), which is consistent with the reported results of a mainly CO product from Si/TiO$_2$-based electrodes (see Supplementary Table S5). After the

deposition of trzpOs catalyst, a main product of CH$_4$ is detected with a high FE$_{CH4}$ of 91.8 ± 3.1%, while FE$_{CO}$ and FE$_{H2}$ are only 6.5 ± 3.5% and 1.7 ± 1.8%, respectively (Fig. 3c). Besides, the Si/TiO$_2$/przpOs also show a high FE$_{CH4}$ of 89.8 ± 3.0%, and low FE$_{CO}$ and FE$_{H2}$ of 8.6 ± 3.1% and 1.5 ± 1.3%, respectively (Fig. 3c). These results indicate that [Os] complexes are efficient CO$_2$-to-CH$_4$ conversion catalysts. In $^{13}$CO$_2$ isotope experiments, $^{13}$CH$_4$ and $^{13}$CO products with m/z of 17 and 29 are detected for the Si/TiO$_2$/trzpOs at 0.0 V$_{RHE}$ in $^{13}$CO$_2$ atmosphere, in 0.5 M Na$_2$SO$_4$ solution, and under illumination (Fig. 3d). In contrast, CH$_4$ and CO products with m/z of 16 and 28 are detected in a CO$_2$ atmosphere at the same conditions. Gas chromatograph-mass spectrometry (GC-MS) results determine that CH$_4$ and CO products are generated from the reduction of CO$_2$ molecules. To the best of our knowledge, this is the first report of PEC systems using [Os] complex as CO$_2$R catalyst, which exhibits ultra-high activity and CH$_4$ product selectivity, exceeding all previously reported Si-based photocathode for CO$_2$R (see Supplementary Table S5).

## Stability of photocathode and [Os] catalyst

After the PEC tests, the cross-sectional structure of Si/TiO$_2$/przpOs and Si/TiO$_2$/trzpOs electrodes were characterized using SEM images and the related elemental mappings. As illustrated in Fig. S8, the post-PEC CO$_2$ reduction displayed a distinct TiO$_2$ protection layer and an [Os] catalyst layer, signifying the structural stability of the photoelectrodes. The XRD pattern exhibited characteristic diffraction peaks of TiO$_2$ and Si were observed, while no distinctive diffraction signals was observed

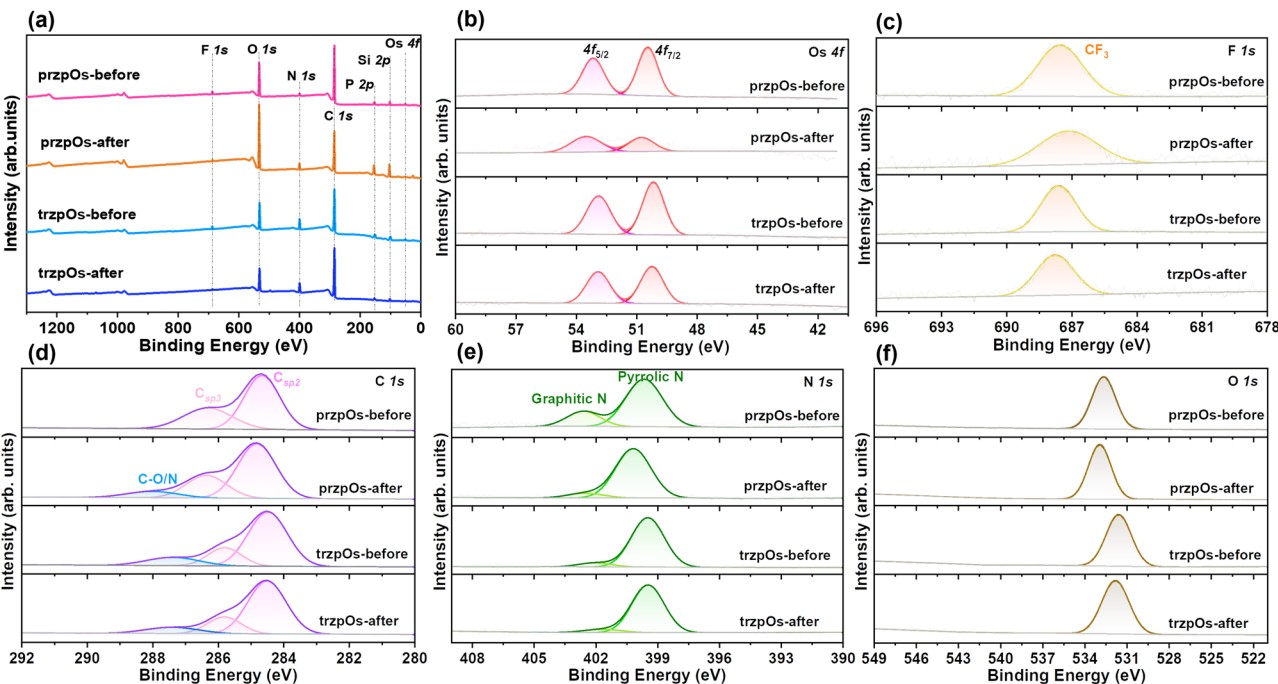

**Fig. 4 | Elemental composition and oxidation state of the electrode before and after PEC test.** XPS survey spectra (**a**) and high-resolution XPS spectra of Os $4f$ (**b**), F $1s$ (**c**), C $1s$ (**d**), N $1s$ (**e**) and O $1s$ (**f**) for Si/TiO$_2$/przpOs and Si/TiO$_2$/trzpOs before and after PEC measurements.

for the [Os] catalysts due to their low loading. Notably, after the PEC test, there were no significant changes in the diffraction peaks of both the electrodes (see Supplementary Fig. S9a). Furthermore, the [Os] catalysts deposited on the Si/TiO$_2$ electrodes were eluted by acetonitrile, and UV-vis spectroscopic studies indicated that the solution contained przpOs and trzpOs (see Supplementary Fig. S9b, c). These experimental findings confirmed that the [Os] complex remained stable on the Si- electrode.

X-ray photoelectron spectroscopy (XPS) was used to compare the element composition and electronic states of Si/TiO$_2$/[Os] electrode before and after PEC CO$_2$R measurements. In Fig. 4a, XPS survey spectra verified the existence of elemental Os, F, P, N, C, O, Si in Si/TiO$_2$/przpOs and Si/TiO$_2$/trzpOs electrodes. In high-resolution XPS spectra, przpOs shows two peaks at 50.4 and 53.2 eV, corresponding to Os $4f_{7/2}$ and $4f_{5/2}$ spin-orbit levels, respectively (Fig. 4b). While, trzpOs displays lower binding energies for its $4f_{7/2}$ and $4f_{5/2}$ peaks, measured at 50.2 eV and 52.9 eV, respectively. The binding energy shift is attributed to the enhanced electron-donating capacity of the triazole ligand. It is worth noting that the binding energy of the [Os] complex deposited on the silicon electrode differs from that of electrodeposited Os metal reported in the literature[59]. Fig. 4c presents the high-resolution XPS spectra of F $1s$, where the peak at 687.6 eV is attributed to the spin-orbit of F $1s$ in CF$_3$ group. Furthermore, Fig. 4d–f exhibit the high-resolution XPS spectra of C $1s$, N $1s$, and O $1s$, respectively, confirming the existence of C, N, O elements from the [Os] complexes. Following PEC measurements, the Si/TiO$_2$/trzpOs electrode demonstrates comparable binding energies and peak intensities for Os, F, C, N, and O elements (Fig. 4b–f). These results indicate that the trzpOs complex exhibits excellent tolerance for CO$_2$ reduction.

## Catalytic mechanism
Density functional theory (DFT) calculations were performed to investigate the structure-property relationship. The absorption spectra and photoluminescence spectra of przpOs and trzpOs were simulated. As shown in Fig. S10, przpOs and trzpOs exhibit maximum absorbance centered ($\lambda_{abs}$) at 423 and 393 nm, respectively, with corresponding maximum emission peaks at 845 and 821 nm. These

calculated results align well with the experimental spectra, indicating that trzpOs possesses a wider bandgap than przpOs. Energy level calculations depicted in Fig. S11 reveal that przpOs and trzpOs have LUMO values of −2.16 eV and −2.39 eV, and HOMO values of −4.67 eV and −4.99 eV, respectively. Furthermore, trzpOs demonstrates a larger dipole moment (μ) of 18.23 D compared to 12.77 D for przpOs.

The frontier orbital distributions of the ground (S$_0$) and excited state (T$_1$) of the Os complexes are explored by DFT calculations (see Supplementary Fig. S12). In the S$_0$ state, the HOMO orbits of przpOs and trzpOs predominantly reside on the 3,3′-bipyrazole and 3,3′-bi(1,2,4-triazole) ligands, while the LUMO orbits are predominantly situated on their phenanthroline ligands. Notably, trzpOs exhibits extensively delocalized HOMO compared to przpOs. Upon excitation, an electron transitions from the original HOMO orbit to the original LUMO orbit, maintaining the LUMO orbital distribution in the excited state, while the HOMO orbital becomes further dispersed. Remarkably, in both the ground and excited states, trzpOs displays more dispersed HOMO orbitals than przpOs, with the same N atom at position 4 exhibiting a distinct electron cloud distribution, making it a favorable binding site for CO$_2$ reactions. Furthermore, electrostatic potential distributions reveal negative charge accumulations on the same N$_4$ atom of przpOs and trzpOs (see Supplementary Fig. S12), indicating a conducive site for the nucleophilic attack reaction with CO$_2$ substrate.

The conversion pathway of CO$_2$ initial from the ground state (S$_0$) or triplet excited state (T$_1$) of przpOs and trzpOs photocatalysts was further studied by DFT calculations[18]. As shown in Figs. S13–S20, the optimized adsorption configurations of [Os] complexes are shown for each intermediate, such as *CO$_2$, *COOH, *CO, *CHO, *CH$_2$O, *CH$_3$, and *CH$_4$, with C atoms binding to the electron deficient N atoms on the bipyrazole and triazole ligands of przpOs and trzpOs, respectively. Notably, when CO$_2$ adsorption occurs in the ground state of przpOs and trzpOs, the Gibbs free energy changes (ΔG) are +0.20 and +0.08 eV, respectively. Subsequently, converting the adsorbed *CO$_2$ in ground state (S$_0$) to *COOH is an endothermic reaction that must overcome an energy barrier of +2.03 and 1.91 eV for przpOs and trzpOs catalysts, respectively (Fig. 5c). Oppositely, when [Os] complex is in T$_1$ state, according to its long luminescence lifetime as shown in Fig. 1d,

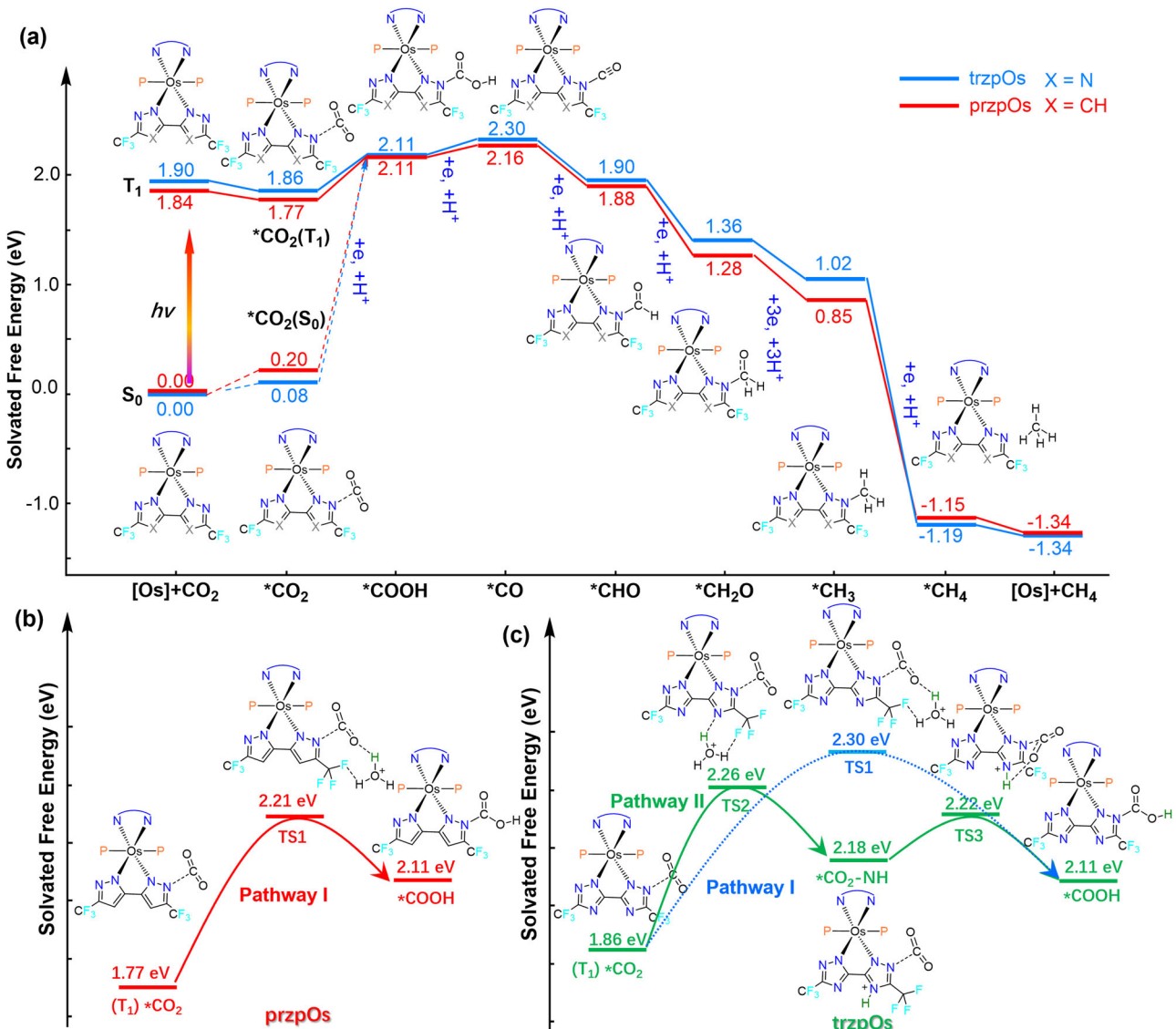

**Fig. 5 | DFT calculations of the CO₂ conversion pathway and the transition states of *CO₂ protonation.** Calculated adsorption configurations of CO₂ and reactive intermediates on przpOs (**a**) and trzpOs (**b**). **a** Gibbs free energy diagrams for CO₂ adsorption and CO₂ reduction to CH₄ by przpOs and trzpOs catalysts. Transition states of przpOs (**b**) and trzpOs (**c**) via directly protonation (Pathway I) and via nitrogen heteroatom ferrying (Pathway II).

the Gibbs free energy for CO₂ adsorption decreases to −0.07 and −0.04 eV for przpOs and trzpOs, respectively. Correspondingly, the required energy for the conversion of *CO₂ to *COOH is dramatically decreased to +0.34 and +0.25 eV, respectively. These results suggest that the formation of *COOH intermediates is the rate-limiting step for further hydrogenation processes[60], and [Os] complexes tend to interact with CO₂ substrates through excited states and drive the subsequent conversion process.

The subsequent CO₂ hydrogenations, for instance, the kinetic protonation process of adsorbed *CO₂, are the crucial steps of the CO₂R reaction. Since the CO₂R reaction was carried out in aqueous, hydrated proton is used as the proton source for the converting *CO₂ to *COOH. As shown in Figs. 5b and S21, przpOs adopts a directly protonation of *CO₂ (Pathway I) with a ΔG of +0.44 eV. In contrast, trzpOs can protonate the N atom on triazole ligand first and then transfer protons to *CO₂, which is a two-step protonation via nitrogen heteroatom ferrying (Pathway II). The transition state energy barrier of two-step protonation of trzpOs is only +0.40 and +0.04 eV, which is more favorable than the direct protonation process (Figs. 5c, S22). These results confirms that the extra nitrogen atom on triazole ligand could serve as a proton-relay to

facilitate the proton transfer for the sequential CO₂ hydrogenations. Afterwards, the intermediate exhibits a downhill free energy change at present of protons and electrons, e.g., the transformation of *CO to *CH₄ and the release of CH₄ are thermodynamically exothermic reactions which can be spontaneously carried out by receiving photogenerated electrons from the Si photocathode (Fig. 5a). DFT results indicate that CH₄ is the most easily desorbed product from the [Os] complexes, resulting in the high selectivity for CO₂-to-CH₄ conversion. In contrast, the required energy for the hydrogenation process from *COOH to *CH₃ of trzpOs is smaller than that of przpOs.

Mott-Schottky plots of the Si, Si/TiO₂ and Si/TiO₂/[Os] electrodes exhibit a negative slope in the potential window of 0.2–0.6 V$_{RHE}$ due to the p-type Si semiconductors (see Supplementary Fig. S23). By extrapolating the linearly fitted lines of these plots to 0 of 1/C₂, the flat potential (E$_{fb}$) is obtained from the intercept. Compared with the E$_{fb}$ of 0.40 V$_{RHE}$ for the bare black Si electrode, the E$_{fb}$ of Si/TiO₂ increases to 0.55 V$_{RHE}$ due to the formation of p-n heterojunction. Additionally, Si/TiO₂/przpOs and Si/TiO₂/trzpOs exhibit positive-shifted E$_{fb}$ of 0.58 and 0.65 V$_{RHE}$ (see Supplementary Fig. S23), respectively, indicating that the deposited [Os] complex can effectively gather charge from the Si

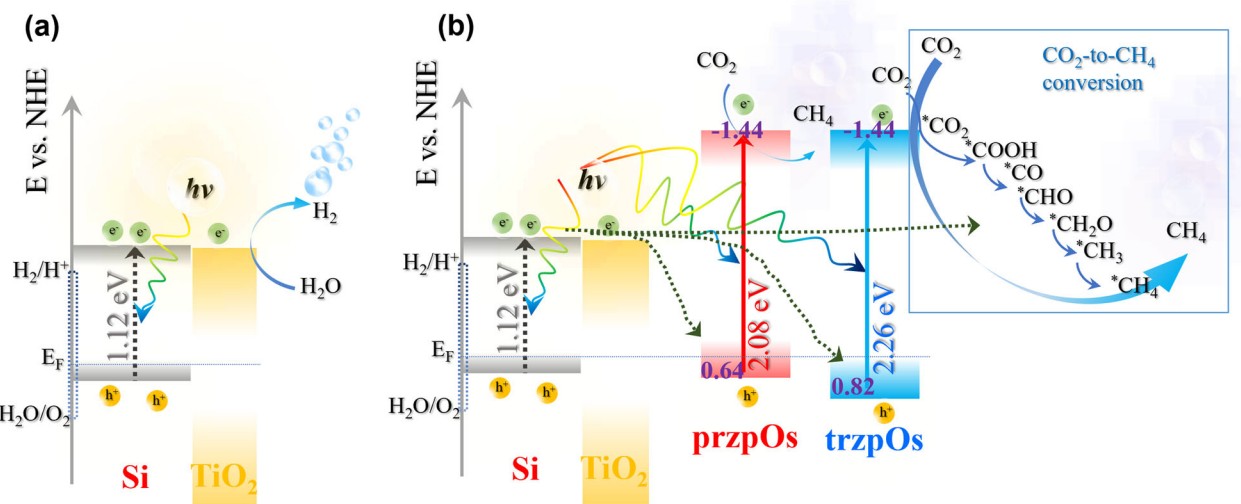

**Fig. 6 | Schematic diagram of energy levels of Si-based photocathode.** Solar-driven $CO_2$-to-$CH_4$ conversion on the Si/$TiO_2$ electrode in the absence (**a**) and presence (**b**) of [Os] complex.

electrode, ensuring efficient charge separation. Moreover, analysis of XPS valence spectra reveals a 0.033 and 0.404 eV difference of valence band (VB) and Fermi level ($E_F$-$E_F$) for Si/$TiO_2$/przpOs and Si/$TiO_2$/trzpOs, respectively (see Supplementary Fig. S24). The difference between $E_{fb}$ and $E_F$ leads to a variation trend of conduction band (CB) and VB levels at the space charge region. Conclusively, the energy level structure diagram of the Si/$TiO_2$/[Os] material was drawn by combining the results of electrochemistry and spectroscopy. Notably, in the absence of [Os] catalysts, the HER predominates at the electrode using electrons from the Si/$TiO_2$ photocathode (Fig. 6a).

In the conversion of $CO_2$ to methane, multiple electron and proton transfer steps involving 8 electrons are necessary. Consequently, the [Os] catalyst must continuously accept electrons from the Si semiconductor. DFT results indicated that the initial reduction of $^*CO_2$ to $^*COOH$ is an energy-consuming process and serves as the rate-determining step for $CO_2$ reduction[60]. Leveraging the [Os] complexes as photocatalyst, the excited [Os] catalysts can lower the energy barrier required for the conversion of $^*CO_2$ to $^*COOH$, facilitating the crucial first step of the $CO_2$ reduction reaction. As depicted in Fig. 6, the Si/$TiO_2$/[Os] displays a Z-scheme heterojunction between Si/$TiO_2$ semiconductor and [Os] complexes, which enables the $CO_2R$ at the LUMO. The HOMO of [Os] complex accepts the photogenerated electrons from the CBs of Si and $TiO_2$[61]. Based on the energy level heterojunction structure, trzpOs exhibits a higher propensity to acquire electrons from Si/$TiO_2$ compared to przpOs, aligning with catalytic activity findings. As the subsequent conversion processes are thermodynamically energy-neutral and the [Os] catalyst intermediates lack sufficient time to capture photons for excited state generation, the [Os] catalyst only accepts electrons from the Si electrode during the intermediate conversion stages rather than functioning as a photosensitizer. Consequently, the generated [Os]-COOH adducts continue to receive electrons from the Si-based electrode, propelling subsequent conversion steps until methane release occurs.

To further understand the interfacial charge and mass transfer processes of Si-based photocathode with [Os] catalysts, electrochemical impedance spectroscopy (EIS) measurements were performed under realistic catalysis conditions with the results presented as Nyquist plots (see Supplementary Fig. S25). Si/$TiO_2$/[Os] demonstrated reduced charge-transfer resistance ($R_{ct}$) and larger interfacial electrochemical double-layer capacitance ($C_{dl}$) in comparison with Si/$TiO_2$ photocathode (Table S6). $C_{dl}$ represents a quantitative parameter to elucidate charge accumulation at the interface. This result indicates the importance of

[Os] complexes in accumulating electrons from semiconductor and improving charge transfer at the electrode-electrolyte interface. Furthermore, the Si/$TiO_2$ electrode functionalized with trzpOs exhibited a lower $R_{ct}$ than its przpOs counterpart, aligning with its superior PEC $CO_2R$ catalytic efficiency.

## Discussion

In conclusion, we introduced two Os metal complexes, namely przpOs and trzpOs, as efficient $CO_2$-to-$CH_4$ conversion catalysts, and combined with Si/$TiO_2$ heterojunction photocathode for direct solar-driven $CO_2R$. As results, the trzpOs with bi(1,2,4-triazole) ligand shows higher $CO_2R$ activity than that of przpOs (6.41 vs. 0.544 s$^{-1}$), owing to the intramolecular incorporation effect caused by the proton binding site provided by the N atom on the triazole ligand. The prepared Si/$TiO_2$/trzpOs photocathode has achieved efficient $CO_2$-to-$CH_4$ conversion with a high $j_{ph}$ −25.8 mA/cm$^2$ at −0.3 $V_{RHE}$ and >90% Faradaic Efficiency for $CH_4$ product. DFT calculations revealed that the N atoms on the bipyrazole and triazole ligands were the key active sites, and strongly adsorbed $^*CO_2$ tended to be further hydrogenated to form $CH_4$, leading to their ultrahigh $CH_4$ selectivity. This work clearly demonstrates that [Os] complex is an efficient catalyst for PEC $CO_2$ reduction and has significant importance for highly efficient and selective solar-driven $CO_2$ conversion study.

## Methods

### Materials and instruments

Concentrated hydrochloric acid (HCl), trifluoroacetic acid (TFA), hydrogen peroxide ($H_2O_2$), ammonia ($NH_3·H_2O$, ~25% w/w in water), potassium chloride (KCl), triethylamine (TEA), potassium bromide (KBr), anhydrous sodium sulfate ($Na_2SO_4$), tetrabutylammonium hexafluorophosphate (n-$Bu_4NPF_6$) and sodium hydroxide (HF, 48.0–55.0% w/w in water) were obtained from Shanghai Pedder Medical Technology Co., LTD. Diethylene glycol monomethyl ether (DGME), trimethylamine oxide ($Me_3NO$), dodecacarbonyltriosmium [$Os_3(CO)_{12}$], 1,10-phenanthroline (phen), dimethyl(phenyl)phosphane, Ethyl acetate (EA) and hexane are purchased from the shanghai bidepharm technology Co., LTD. The electrolytes of $Na_2SO_4$ (pH 6.8 ± 0.3) and n-$Bu_4NPF_6$ are electroanalytically purity, while the commonly used chemicals are analytical grade and be used without further purification. 100 μm of boron-doped p-type (100) Si wafer with resistivity of 1–10 Ω cm was purchased from Zhejiang Jingyou Silicon Technology Co., Ltd. The targets of Al and $TiO_2$ used

in the magnetron sputtering technology were purchased from Zhongnuo New Materials Co., LTD.

Field emission scanning electron microscope and the related EDXS morphologies were recorded on a Zeiss Gemini 300 instrument. All X-ray Multifunctional imaging electron spectrometer experiments were obtained on a Thermo ESCALAB 250Xi with a monochromatized Al Kα. Absorption and photoluminescence spectra are collected with a Shimadzu UV2600 UV-Vis spectrophotometer and a Hitachi F7000 luminescence spectrophotometer, respectively. Nuclear magnetic resonance (NMR) spectra were recorded on the Bruker AscendTM 400 MHz (or 600 MHz) NMR spectrometer with tetramethylsilane (TMS) as an internal standard. X-ray diffraction (XRD) was recorded on the D8 ADV ANCE instrument manufactured by Bruker, Germany. XPS was performed by Thermo ESCALAB 250Xi spectroscopy.

## Synthesis of przpOs and trzpOs

General procedure for synthesizing Os complex: A mixture of $Os_3(CO)_{12}$ (150 mg, 0.165 mmol), 1,10-phenanthroline (90 mg, 0.500 mmol) and bpzH$_2$ (135 mg, 0.500 mmol) (or btzH$_2$, 136 mg, 0.500 mmol) in 20 mL of DGME was heated to 185 °C for 48 h. After cooling to room temperature, freshly sublimed Me$_3$NO (78 mg, 1.035 mmol) was added, and the solution was then heated to 185 °C for 2 h. Then, dimethyl(phenyl)phosphane (218 mg, 1.08 mmol) was added, and the solution was heated to 185 °C for another 12 h. Finally, the solvent was removed under vacuum, and the residue was purified by silica gel column chromatography eluting with EA/hexane (4:1) yielding przpOs (353 mg, 78%) and trzpOs (236 mg, 52%) as black solids.

## Fabrication of Si-based photocathode

Si photocathodes were fabricated with ohmic contact of 200 nm Al on the backside. Then, Si wafer was connected to Cu tape, fixed on a plastic plate, and sealed the edges with epoxy resin. The surface engineering of the nanoporous structures and p-n heterojunction were constructed by PEC etching in HF solution[58]. $TiO_2$ thin film was deposited on Si photocathodes by radio frequency magnetron sputtering from the $TiO_2$ (99.99%) target without substrate heating. The thickness of $TiO_2$ film was controlled by the deposition power and time, which were set to 60 W and 40 min to prepare 100 nm $TiO_2$. Finally, Si/$TiO_2$/[Os] photocathodes (0.2 × 0.5 cm$^2$) were prepared by dropping the acetonitrile solution of [Os] complexes (1.0 mM) onto the Si/$TiO_2$ electrode surface and drying at room temperature.

## Electrochemical and photoelectrochemical measurements

Electrochemical and PEC experiments were measured on a CHI760E potentiostat. Aqueous electrolyte of 0.5 M Na$_2$SO$_4$ and acetonitrile solution of 0.1 M n-Bu$_4$NPF$_6$ were prepared in volumetric flasks and stored at room temperature. Cyclic voltammetry test of Os complexes were measured in acetonitrile solvent containing 0.1 M n-Bu$_4$NPF$_6$ electrolyte, using Glassy Carbon, Ag/Ag$^+$ (0.1 M Ag$^+$) and 1 × 1 cm$^2$ Pt plate as working, reference, and counter electrodes, respectively. PEC experiments were measured in aqueous electrolyte, in a three-electrode system consisting of the prepared Si photocathode, the counter electrode of Pt plate and the reference electrode Ag/AgCl (saturated KCl). The simulated sunlight of AM1.5 G (100 mW cm$^{-2}$) was supplied from the solar simulator (China Education Au-Light Co., Ltd). The EIS of Si/$TiO_2$, Si/$TiO_2$/przpOs and Si/$TiO_2$/trzpOs electrodes were measured in $CO_2$-saturated 0.5 M Na$_2$SO$_4$ under illumination in the frequency range of 1–10$^6$ Hz. The potential measured with respect to Ag/AgCl (V$_{Ag/AgCl}$) was converted to the potential versus reversible hydrogen electrode (V$_{RHE}$) using the following equation: V$_{RHE}$ = V$_{Ag/AgCl}$ + E$_0$ + 0.059 × pH, where E$_0$ is the potential of the Ag/AgCl reference electrode with respect to the standard hydrogen potential (V$_{NHE}$). The reference electrodes of Ag/Ag$^+$ (0.1 M Ag$^+$) and Ag/AgCl (saturated KCl) are used directly after purchase from GaossUnion company. The linear sweep voltammetry (LSV) curves were measured at a scanning rate of 30 mV/s$^{-1}$.

Gas products were monitored by gas chromatography (GC) analysis on an Aulight (GC7920) instrument. The $CO_2$ gas was continuously blown into the PEC cell at a flow rate of 30 sccm, and the sample was automatically injected. Gas chromatography has two FIDs to detect hydrocarbon and CO products, and TCD to detect hydrogen products. Gas products of H$_2$, CO and CH$_4$ were detected for these Si-based photocathodes. The FE of gaseous products were given by the equation: FE = n$_e$ × x × F × flow rate × P ÷ R ÷ T ÷ I, where n$_e$ is the number of moles of electrons required to obtain 1 molecule of product, x is the ppm of gaseous product detected by GC, P is ambient pressure 101325 Pa, T is ambient temperature, F is the Faraday constant (96485 C mol$^{-1}$), R is gas constant (8.314 J K$^{-1}$ mol$^{-1}$), and I is average current density during sampling time.

## Theoretical calculations

All geometric optimizations have been carried out by DFT using the PBE0 hybrid functional[62] with Grimme's dispersion correction of D3 version[63] implemented in Gaussian 16 suite of programs[64]. The standard 6–31 G(d,p) basis set[65–67] for H, C, N, O, and P atoms was used. For the Os atom, the LANL2DZ basis set and its corresponding effective core potential[68] was used. Frequency calculations at the same level of theory have also been performed to identify all stationary points as minima (zero imaginary frequencies). The single-point energy (SP) calculations were performed on the optimized geometries with def2-TZVP basis set[69,70]. Approximate solvent effects were also taken into consideration based on the continuum solvation model in optimization[71] and SP calculations[72]. The triplet excited states were calculated by modifying the spin multiplicity. The energy profiles of the elementary steps are based on CHE model[73]. Frequency outcomes were examined to confirm stationary points as minima (no imaginary frequencies) or transition states (only one imaginary frequency).

## Data availability

The data supporting the findings of this work are available within the article and its Supplementary Information files. All the data reported in this work are available from the authors. Source data are provided with this paper.

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

## Acknowledgements

Q.X.T. gratefully acknowledges financial support from the National Natural Science Foundation of China (No. 52273187 and 51973107), Guangdong Province Universities and Colleges Pearl River Scholar Funded Scheme 2019 (GDUPS 2019), and Guangdong Basic and Applied Basic Research Foundation (2023A1515240077). J.X.J. Jian acknowledges financial supports from the start-up fund from Shantou University (NTF20033), Guangdong Basic and Applied Basic Research Foundation (2022A1515110372, 2023A1515011306).

## Author contributions

X.-Y. Li: Data curation, Methodology, Formal analysis. Z.-L. Zhu: Methodology, Formal analysis, Resources. F. W. Dagnaw: Writing - original draft. J.-R. Yu: DFT calculation, Software. Z.-X. Wu: Reviewing and editing, Formal analysis. Y.-J. Chen: Data curation, Methodology. M.-H. Zhou: Data curation. T. Wang: Resources. Q.-X. Tong: Supervision, Reviewing and editing, Project administration. J.-X. Jian: Supervision, Reviewing and editing, Project administration.

## Competing interests

The authors declare no competing interests.
