## [Peer Review File · Nature Communications]

REVIEWER COMMENTS

Reviewer #1 (Remarks to the Author):

In this study, two types of synthesized osmium complexes have been developed and introduced as cocatalysts for the Si photocathode in the photoelectrochemical CO₂ reduction reaction. The resulting Si/TiO₂/[Os] complexes exhibited exceptional performance, achieving over 90% Faradaic efficiency in the selective synthesis of methane. This finding is of utmost significance as there is currently no paper demonstrating selective methane production from CO₂. This seminal result significantly advances our understanding of catalytic processes for CO₂ reduction. With some necessary revisions, this research shows promise for acceptance in Nature Communications.

1. When employing organometallic catalysts for CO₂ reduction, the catalyst undergoes reduction to metal on the electrode surface, while the ligand influences CO₂ activity through homogeneous reactions in the electrolyte. It is observed that changing the electrolyte post-reaction leads to decreased activity, but reintroducing the ligand into the electrolyte enhances activity again. It is imperative to investigate whether the ligand of Os complexes exhibits activity on the electrode surface or acts as a homogeneous catalyst in the electrolyte by replacing the electrolyte.

2. The structure of Os complexes should be confirmed post-PEC CO₂ reduction. Incorporating XRD analysis (Si/TiO₂/Os) before and after the CO₂ reduction reaction is necessary to ascertain whether they remain as Os complexes or transform into metal particles. It is crucial to confirm the presence of any Os metals on the Si/TiO₂ surface. Moreover, the observation of large photocurrents of Si/TiO₂/przpOs and Si/TiO₂/trzpOs in N₂ conditions suggests promising co-catalytic activity for hydrogen generation. It is also needed to investigate their stability for this reaction and determine whether they retain their identity as Os complexes or transition into metal particles.

3. It is anticipated that the elevated pH resulting from OH⁻ generation during CO₂ reduction on Si photocathodes may lead to structural damage, even in the presence of a TiO₂ protection layer. Therefore, comparing the Si/TiO₂/Os structure through SEM cross-sectional images before and after the CO₂ reduction reaction is crucial for understanding this phenomenon.

4. If the reaction follows a Z-scheme mechanism, poor photosensitizer performance may limit overall activity. In this scenario, Os complexes with a large band-gap compared to Si photocathode could potentially restrict photocurrent. However, the observed photocurrent behavior resembles that of a Si photocathode, suggesting co-catalytic behavior of the Os complexes. Further discussions are needed to elucidate these mechanisms.

5. Several typos have been identified:

In Line 53, 'CO4-to-CO' should be revised to 'CO2-to-CO'.

In Line 222, '3CO2' should be revised to '13CO2'.

Table S4 requires revision, as it appears to be distorted in the PDF version.

Reviewer #2 (Remarks to the Author):

In this work, the authors designed two Os complex przpOs and trzpOs on Si/TiO₂, which showed high photocatalytic performance to converse CO₂ to CH₄ with FE over 90%. However, from my perspective, the explanation of mechanism based on DFT calculations is not solid enough and couldn't match the level of Nature Communications. Here are some comments for computational section:

1.The computational method should be exhibited in detail. The energy profiles of the elementary steps seem to be based on CHE model, and so the elementary reaction employed should be shown and the crucial intermediate *CO missed in current graph. The method for triplet excited state computations should also be shown, TDDFT or just modifying spin multiplicity.

2.Compared with przpOs, the 3,3'-bi(1,2,4-triazole) ligand in trzpOs possessed a N atom in environment different from 3,3'-bipyrazole in przpOs. Thus, this N atom should also be considered as a possible active site.

3.The energy barriers of *CO₂ to *COOH of S₀ state were close to the excitation energy. So if the decrease of *CO₂/*COOH barriers of T₁ state could be ascribed to the returning to the ground state from excited state?

4.Figure S11 showed the LUMO and HUMO orbits of ground-state przpOs and trzpOs. However, in Figure 5, the authors explained that the high catalytic performance was originated from the excited state. Thus, the difference of the properties between ground and excited states should be further investigated to clearly explain the mechanism of the high performance.

5.The free energy change of whole CRR should be same on different catalysts. The final energy difference (CH₄) may be owing to the structure distortion of przpOs and trzpOs, which is unfavorable for a stable catalyst. The author should check it.

6.In Figure 5, there is ignorable interaction between CO₂ and przpOs/trzpOs can be observed corresponding with the small adsorption energy. Thus, more computational techniques should be employed here to validate authors' clarification "which is conducive to binding CO₂ and protons and achieving high CO₂R activity" (Page 16, Line 290-191)

7. Besides, the optical properties of przpOs and trzpOs, absorption spectrum and fluorescence spectrum for example, should be computed and analyzed in computational section.

8. Page 17, Line 304-305, CHE model is not sufficient to determine the rate-limiting step because it's a thermodynamic energy profile. The authors should perform transition state computation to obtain the dynamic barrier of each elementary steps with explicit solvent model (for better description of proton).

9. The text contains some grammar errors. The author should be more careful.

Reviewer #3 (Remarks to the Author):

The author proposed an interesting photocathode for solar-driven CO₂-to-CH₄ conversion by coupling [Os] complexes with a widely available silicon. The photocathode demonstrated a considerable photocurrent density of -14.11 mA cm⁻² at 0.0 VRHE with a high CH₄ selectivity of >90%. The reaction mechanism was theoretically and spectroscopically studied. However, considering the top level of Nature Communications, this work can't be accepted before the following issues are addressed. Please see below the detailed comments.

1. The underlying cause of the correlation of different catalytic rate constants between PrzpOs and trzp Os with their performance variation needs to be elaborated with evidence.
2. The authors claimed that the successful preparation of przpOs and trzpOs were confirmed by various techniques. It is suggested to elaborate the relative discussions.
3. It is suggested to discuss the correlation of UV-Vis absorption of both przpOs and trzpOs with the activity in Page 6.
4. In page 7, the catalytic current enhancement of trzpOs may be due to a broad range of reasons. It is not scientific enough to attribute the enhancement merely to that the N atoms on the triazole ligand provide binding site for protons to promote the PECT process.
5. There is lack of evidence for CO₂ adduct, which may be a key intermediate of CO₂-to-CH₄ conversion.
6. The writing needs polishment.

Reviewer #1 (Remarks to the Author):

In this study, two types of synthesized osmium complexes have been developed and introduced as cocatalysts for the Si photocathode in the photoelectrochemical CO₂ reduction reaction. The resulting Si/TiO₂/[Os] complexes exhibited exceptional performance, achieving over 90% Faradaic efficiency in the selective synthesis of methane. This finding is of utmost significance as there is currently no paper demonstrating selective methane production from CO₂. This seminal result significantly advances our understanding of catalytic processes for CO₂ reduction. With some necessary revisions, this research shows promise for acceptance in Nature Communications.

1. When employing organometallic catalysts for CO₂ reduction, the catalyst undergoes reduction to metal on the electrode surface, while the ligand influences CO₂ activity through homogeneous reactions in the electrolyte. It is observed that changing the electrolyte post-reaction leads to decreased activity, but reintroducing the ligand into the electrolyte enhances activity again. It is imperative to investigate whether the ligand of Os complexes exhibits activity on the electrode surface or acts as a homogeneous catalyst in the electrolyte by replacing the electrolyte.

Response: Thanks a lot for your helpful advice.

In high-resolution XPS spectra, przpOs shows two peaks at 50.4 and 53.2 eV, corresponding to Os 4f_{7/2} and 4f_{5/2} spin-orbit levels, respectively (**Figure 4b**). While, trzpOs displays lower binding energies for its 4f_{7/2} and 4f_{5/2} peaks, measured at 50.2 eV and 52.9 eV, respectively. The binding energy shift is attributed to the enhanced electron-donating capacity of the triazole ligand. It is worth noting that binding energy of the [Os] complex deposited on the silicon electrode differs from that of electrodeposited Os metal reported in the literature. (Rhee, C. K. *et al.* Osmium nanoislands spontaneously deposited on a Pt(111) electrode: an XPS, STM and GIF-XAS study. *J. Electroanal. Chem.* **554-555**, 367.). Following PEC measurements, the Si/TiO₂/trzpOs electrode demonstrates comparable binding energies and peak intensities for Os, F, C, N, and O elements (**Figure 4b-f**). These results indicate that the trzpOs complex exhibits excellent tolerance for CO₂ reduction.

Due to the conjugated structure of the organic ligands (bpzH₂ and btzH₂), they have obvious absorption signals in acetonitrile solution (Figure 1Rb). After the PEC measurements, no obvious organic ligand signal was observed in the absorption spectrum of the electrolyte solution, indicating that the [Os] complex was stable.

In order to verify whether the organic ligand acts as a homogeneous catalyst, 1.0 mM organic ligands were added to the electrolyte during the LSV measurements using the Si/TiO₂ electrode. As shown in Figure R1c, the photocurrents of the Si/TiO₂ electrode increase in the range of 0.6 to -0.3 V_{RHE} in the electrolyte solution with bpzH₂ or btzH₂. However, the current of the Si/TiO₂ electrode in the organic ligand solution is not significantly different in N₂ and CO₂ atmospheres, indicating that bpzH₂ and btzH₂ ligands are not efficient homogeneous catalysts for CO₂ reduction. In addition, the loading amount of [Os] catalysts on Si/TiO₂ photocathode is as low as 1.6 nmol/cm², and no organic ligands are detected in the electrolyte solutions after PEC measurements. Therefore, the contribution of homogeneous catalysis of organic ligands is extremely low and can be ignored.

Figure R1. (a) LSV curves performed by Si/TiO₂, Si/TiO₂/przpOs and Si/TiO₂/trzpOs electrode in N₂ and CO₂ atmosphere. (b) UV-vis absorption spectra of bpzH₂ and bptH₂ in CH₃CN, and the electrolytes of Si/TiO₂/przpOs and Si/TiO₂/trzpOs after PEC measurements. (c) LSV curves of Si/TiO₂ in 0.1 M KHCO₃ solution without and with adding 1.0 mM of bpzH₂ and btzH₂, in N₂ and CO₂ atmosphere.

2. The structure of Os complexes should be confirmed post-PEC CO₂ reduction. Incorporating XRD analysis (Si/TiO₂/Os) before and after the CO₂ reduction reaction is necessary to ascertain whether they remain as Os complexes or transform into metal

particles. It is crucial to confirm the presence of any Os metals on the Si/TiO₂ surface. Moreover, the observation of large photocurrents of Si/TiO₂/przpOs and Si/TiO₂/trzpOs in N₂ conditions suggests promising co-catalytic activity for hydrogen generation. It is also needed to investigate their stability for this reaction and determine whether they retain their identity as Os complexes or transition into metal particles.

Response: Thanks a lot for your useful advice.

After the PEC tests, the cross-sectional structure of Si/TiO₂/przpOs and Si/TiO₂/trzpOs electrodes were characterized using SEM images and the related elemental mappings. As illustrated in **Figure S8**, the post-PEC CO₂ reduction displayed a distinct TiO₂ protection layer and an [Os] catalyst layer, signifying the structural stability of the photoelectrodes. The XRD pattern exhibited characteristic diffraction peaks of TiO₂ and Si were observed, while no distinctive diffraction signals were observed for the [Os] catalysts due to their low loading. Notably, after the PEC test, there were no significant changes in the diffraction peaks of both the electrodes (**Figure S9a**). Furthermore, the [Os] catalysts deposited on the Si/TiO₂ electrodes were eluted by acetonitrile, and UV-vis spectroscopic studies indicated that the solution contained przpOs and trzpOs (**Figure S9b-c**). These experimental findings confirmed that the [Os] complex remained stable on the Si- electrode.

Figure S9. Stability of electrodes and catalysts. (a) XRD pattern of Si/TiO₂/przpOs and Si/TiO₂/trzpOs electrodes before and after PEC tests. (b) UV-vis absorption spectra of przpOs solution and after PEC test. (c) UV-vis absorption spectra of trzpOs solution and after PEC tests. Inset figures show the preparation process of the [Os] sample after the elution from the electrode after PEC tests.

3. It is anticipated that the elevated pH resulting from OH⁻ generation during CO₂ reduction on Si photocathodes may lead to structural damage, even in the presence of a TiO₂ protection layer. Therefore, comparing the Si/TiO₂/Os structure through SEM cross-sectional images before and after the CO₂ reduction reaction is crucial for understanding this phenomenon.

Response: Thanks a lot for your useful advice.

After the PEC tests, the cross-sectional structure of Si/TiO₂/przpOs and Si/TiO₂/trzpOs electrodes were characterized using SEM images and the related elemental mappings. As illustrated in **Figure S8**, the post-PEC CO₂ reduction displayed a distinct TiO₂ protection layer and an [Os] catalyst layer, signifying the structural stability of the photoelectrodes.

Figure S8. The element distribution of electrodes after PEC test. (a) Cross-sectional SEM images of Si/TiO₂/przpOs after PEC test, and the related elemental mapping of Si (b), Ti (c), Os (d), F (e) and Os (f). (g) Cross-sectional SEM images of Si/TiO₂/trzpOs after PEC test, and the related elemental mapping of Si (h), Ti (i), Os (j), F (k) and Os (l).

Besides, Si-based photocathode was stable in electrochemical tests, which has been confirmed in our previous work (*Catal. Sci. Technol.* **2022**, *12*, 5640.) and other literature. However, when Si is used in a photoanode, its stability is worth worrying due to the problem of anode corrosion, and an effective protective layer such as TiO₂ is usually required.

4. If the reaction follows a Z-scheme mechanism, poor photosensitizer performance may limit overall activity. In this scenario, Os complexes with a large band-gap compared to Si photocathode could potentially restrict photocurrent. However, the observed photocurrent behavior resembles that of a Si photocathode, suggesting co-catalytic behavior of the Os complexes. Further discussions are needed to elucidate these mechanisms.

Response: Thanks a lot for your useful advice.

We rearranged the means of expression in the context, that is, we first analyzed the DFT calculation results, and then discussed the charge-transfer mechanism between Si-based semiconductors and [Os] catalysts in details.

In the conversion of CO₂ to methane, multiple electron and proton transfer steps involving 8 electrons are necessary. Consequently, the [Os] catalyst must continuously accept electrons from the Si semiconductor. DFT results indicated that the initial reduction of *CO₂ to *COOH is an energy-consuming process and serves as the rate-determining step for CO₂ reduction.⁶⁰ Leveraging the [Os] complexes as photocatalyst, the excited [Os] catalysts can lower the energy barrier required for the conversion of *CO₂ to *COOH, facilitating the crucial first step of the CO₂ reduction reaction. As depicted in **Figure 6**, the Si/TiO₂/[Os] displays a Z-scheme heterojunction between Si/TiO₂ semiconductor and [Os] complexes, which enables the CO₂R at the LUMO. The HOMO of [Os] complex accepts the photogenerated electrons from the CBs of Si and TiO₂.⁶¹ Based on the energy level heterojunction structure, trzpoOs exhibits a higher propensity to acquire electrons from Si/TiO₂ compared to przpoOs, aligning with catalytic activity findings. As the subsequent conversion processes are thermodynamically energy-neutral and the [Os] catalyst intermediates lack sufficient

time to capture photons for excited state generation, the [Os] catalyst only accepts electrons from the Si electrode during the intermediate conversion stages rather than functioning as a photosensitizer. Consequently, the generated [Os]-COOH adducts continue to receive electrons from the Si-based electrode, propelling subsequent conversion steps until methane release occurs.

Figure 6. Schematic diagram of energy levels of Si-based photocathode. Solar-driven CO_2 -to- CH_4 conversion on the Si/ TiO_2 electrode in the absence (a) and presence (b) of [Os] complex.

5. Several typos have been identified:

In Line 53, ‘ CO_4 -to- CO ’ should be revised to ‘ CO_2 -to- CO ’.

In Line 222, ‘ $3CO_2$ ’ should be revised to ‘ $^{13}CO_2$ ’.

Table S4 requires revision, as it appears to be distorted in the PDF version.

Response: Thanks a lot for your useful advice.

We have made corresponding corrections in the revised draft. We also checked the grammar and spelling of the article thoroughly.

Reviewer #2 (Remarks to the Author):

In this work, the authors designed two Os complex przpOs and trzpOs on Si/TiO₂, which showed high photocatalytic performance to converse CO₂ to CH₄ with FE over 90%. However, from my perspective, the explanation of mechanism based on DFT calculations is not solid enough and couldn't match the level of Nature Communications. Here are some comments for computational section:

1. The computational method should be exhibited in detail. The energy profiles of the elementary steps seem to be based on CHE model, and so the elementary reaction employed should be shown and the crucial intermediate *CO missed in current graph. The method for triplet excited state computations should also be shown, TDDFT or just modifying spin multiplicity.

Response: Thanks a lot for your useful advice.

The calculation method was supplemented in the Method section:

“All geometric optimizations have been carried out by density functional theory using the PBE0 hybrid functional⁶² with Grimme's dispersion correction of D3 version⁶³ implemented in Gaussian 16 suite of programs⁶⁴. The standard 6-31G(d,p) basis set⁶⁵⁻⁶⁷ for H, C, N, O, and P atoms was used. For the Os atom, the LANL2DZ basis set and its corresponding effective core potential⁶⁸ was used. Frequency calculations at the same level of theory have also been performed to identify all stationary points as minima (zero imaginary frequencies). The single-point energy (SP) calculations were performed on the optimized geometries with def2-TZVP basis set^{69,70}. Approximate solvent effects were also taken into consideration based on the continuum solvation model in optimization⁷¹ and SP calculations⁷². The triplet excited states were calculated by modifying the spin multiplicity. The energy profiles of the elementary steps are based on CHE model⁷³.”

The *CO intermediate is added in the revised Figure 5 and the corresponding description is added.

2. Compared with przpOs, the 3,3'-bi(1,2,4-triazole) ligand in trzpOs possessed a N atom in environment different from 3,3'-bipyrazole in przpOs. Thus, this N atom should

also be considered as a possible active site.

Response: Thanks a lot for your useful advice.

Electrochemical analysis have revealed distinct differences in the $k_{\text{cat-H}}$ values of przpOs and trzpOs catalysts when an additional proton source is introduced. We hypothesized that an extra nitrogen atom on the 3,3'-bi(1,2,4-triazole) ligand in trzpOs serves as a proton-relay that facilitates the proton transfer for the sequential CO_2 reduction process. The N atom can provide binding sites for protons, which was verified in subsequent comparative experiments. Under N_2 atmosphere, as TFA was incrementally added to the acetonitrile solution of przpOs, its E_2 peak became irreversible with a notable increase in the cathodic current, pointing towards a characteristic of the hydrogen evolution process (HER) (**Figure S5**). Conversely, the HER current of trzpOs demonstrated a considerably greater enhancement compared to that of przpOs, thus confirming the superior proton-binding capability and showcasing trzpOs' heightened proton-reduction activity.

Figure S5. Electrochemical HER of [Os] complex. CV curves of 1.0 mM of przpOs (a) and trzpOs (b) in N_2 atmosphere with the addition of proton source of TFA.

The frontier orbital distributions of the ground (S_0) and excited state (T_1) of the Os complexes are explored by DFT calculations (**Figure S12**). In the S_0 state, the HOMO orbits of przpOs and trzpOs predominantly reside on the 3,3'-bipyrazole and 3,3'-bi(1,2,4-triazole) ligands, while the LUMO orbits are predominantly situated on their phenanthroline ligands. Notably, trzpOs exhibits extensively delocalized HOMO compared to przpOs. Upon excitation, an electron transitions from the original HOMO

orbit to the original LUMO orbit, maintaining the LUMO orbital distribution in the excited state, while the HOMO orbital becomes further dispersed. Remarkably, in both the ground and excited states, trzpOs displays more dispersed HOMO orbitals than przpOs, with the same N atom at position 4 exhibiting a distinct electron cloud distribution, making it a favorable binding site for CO₂ reactions. Furthermore, electrostatic potential distributions reveal negative charge accumulations on the same N₄ atom of przpOs and trzpOs (Figure S12), indicating a conducive site for the nucleophilic attack reaction with CO₂ substrate.

Figure S12. The orbital distribution and electrostatic potential of singlet and triplet states. The distributions of HOMO and LUMO of przpOs in S₀ (a) and T₁ (b) states, and trzpOs in S₀ (c) and T₁ (d) states. Electrostatic potential of przpOs in S₀ (e) and T₁ (f) states, and trzpOs in S₀ (g) and T₁ (g) states, respectively.

Intrigued by these fascinating results, currently, we prepared many series of including the [Os] complexes with 2,2'-bipyrrrole and 2,2'-biimidazole ligands to explore the catalytic activity and mechanism of [Os] catalyst with N atoms at the additional N position and will publish in the near future.

3. The energy barriers of *CO_2 to *COOH of S_0 state were close to the excitation energy. So if the decrease of $^*CO_2/^*COOH$ barriers of T_1 state could be ascribed to the returning to the ground state from excited state?

Response: Thanks a lot for your advice.

The $^*CO_2/^*COOH$ energy barrier of the T_1 state is lower than that of the S_0 state, which is attributed to the energy provided by the photons absorbed by the [Os] catalyst. In the photocatalytic process, the catalyst absorbs photons into a high-energy excited state, and then reacts with the substrate to form a catalytic intermediate, which is the characteristic of photocatalysis different from thermal catalysis. In contrast, if the catalytic cycle starts from the ground state, more energy is required to reach the same intermediate. The intermediate species of the catalytic cycle are high-energy active species, and their total number of electrons is different from the ground state or excited state of the catalyst.

4. Figure S11 showed the LUMO and HOMO orbits of ground-state przpOs and trzpOs. However, in Figure 5, the authors explained that the high catalytic performance was originated from the excited state. Thus, the difference of the properties between ground and excited states should be further investigated to clearly explain the mechanism of the high performance.

Response: Thanks a lot for your useful advice.

The frontier orbital distributions of the ground (S_0) and excited state (T_1) of przpOs and trzpOs were investigated by DFT calculations. In the S_0 state, the HOMO orbits of przpOs and trzpOs are mainly located on the 3,3'-bipyrazole and 3,3'-bi(1,2,4-triazole) ligands, while the LUMO orbits are mainly distributed on their phenanthroline ligands. The trzpOs has widely delocalized HOMO orbitals in comparison with przpOs. During the excitation process, an electron from the original HOMO orbit is excited into the original LUMO orbit, so the LUMO orbital distribution of the excited state is maintained, while the HOMO orbital is further dispersed. In contrast, both in the ground and excited states, the HOMO orbitals of trzpOs are more dispersed than those of przpOs, and the same N atom at position 4 has a clear electron cloud distribution, which

is beneficial to serve as a binding site for the CO₂ reaction.

Figure R2. The frontier orbital distributions of HOMO and LUMO of przpOs in S₀ (a) and T₁ (b) states, and trzpOs in S₀ (c) and T₁ (d) states, respectively.

5. The free energy change of whole CRR should be same on different catalysts. The final energy difference (CH₄) may be owing to the structure distortion of przpOs and trzpOs, which is unfavorable for a stable catalyst. The author should check it.

Response: Thanks a lot for your advice.

The free energy change of the whole CRR is the same. In this work, the DFT calculations take the excited state T₁ of the two catalysts as the energy zero point, and the catalysts are restored to the ground state after the catalytic cycle by releasing methane. Due to the difference in the T₁ and S₀ states of przpOs and trzpOs, the energy difference between the final methane products is not the same. Additionally, the simulated luminescence spectra of przpOs and trzpOs have the maximum emission peaks at 845 and 821 nm, confirming that the T₁ to S₀ conversion energy difference of trzpOs is about 0.06 eV higher than that of przpOs. Consequently, the Gibbs free energy change of trzpOs catalyst releasing CH₄ and returning to S₀ is 0.06 eV lower.

6. In Figure 5, there is ignorable interaction between CO₂ and przpOs/trzpOs can be observed corresponding with the small adsorption energy. Thus, more computational techniques should be employed here to validate authors' clarification "which is conducive to binding CO₂ and protons and achieving high CO₂R activity" (Page 16,

Line 290-191)

Response: Thanks a lot for your useful advice.

We added the frontier orbital distributions and the electrostatic potential distributions of the ground (S_0) and excited state (T_1) of przpOs and trzpOs to validate the active sites for CO_2 in Figure S12.

In the S_0 state, the HOMO orbits of przpOs and trzpOs predominantly reside on the 3,3'-bipyrazole and 3,3'-bi(1,2,4-triazole) ligands, while the LUMO orbits are predominantly situated on their phenanthroline ligands. Notably, trzpOs exhibits extensively delocalized HOMO compared to przpOs. Upon excitation, an electron transitions from the original HOMO orbit to the original LUMO orbit, maintaining the LUMO orbital distribution in the excited state, while the HOMO orbital becomes further dispersed. Remarkably, in both the ground and excited states, trzpOs displays more dispersed HOMO orbitals than przpOs, with the same N atom at position 4 exhibiting a distinct electron cloud distribution, making it a favorable binding site for CO_2 reactions. Furthermore, electrostatic potential distributions reveal negative charge accumulations on the same N_4 atom of przpOs and trzpOs (**Figure S12**), indicating a conducive site for the nucleophilic attack reaction with CO_2 substrate.

Figure S12. The orbital distribution and electrostatic potential of singlet and triplet states. The distributions of HOMO and LUMO of przpOs in S_0 (a) and T_1 (b) states, and trzpOs in S_0 (c) and T_1 (d) states. Electrostatic potential of przpOs in S_0 (e) and T_1 (f) states, and trzpOs in S_0 (g) and T_1 (g) states, respectively.

7. Besides, the optical properties of przpOs and trzpOs, absorption spectrum and fluorescence spectrum for example, should be computed and analyzed in computational section.

Response: Thanks a lot for your useful advice.

The absorption spectra and photoluminescence spectra of przpOs and trzpOs were simulated using DFT calculations. As shown in **Figure S10**, przpOs and trzpOs exhibit maximum absorbance centered (λ_{abs}) at 423 and 393 nm, respectively, with corresponding maximum emission peaks at 845 and 821 nm. These calculated results align well with the experimental spectra, indicating that trzpOs possesses a wider bandgap than przpOs.

Figure S10. Simulated absorption and luminescence spectra. DFT calculation of absorption and luminescence spectra of przpOs (a) and trzpOs (b).

	Exp. Abs. λ_{max} (nm)	Cal. Abs. λ_{max} (nm)	Exp. PL λ_{max} (nm)	Cal. PL λ_{max} (nm)
przpOs	485	423	839	845
trzpOs	456	393	783	821

8. Page 17, Line 304-305, CHE model is not sufficient to determine the rate-limiting step because it's a thermodynamic energy profile. The authors should perform transition state computation to obtain the dynamic barrier of each elementary steps with explicit solvent model (for better description of proton).

Response: Thanks a lot for your advice.

The whole CO₂-to-CH₄ conversion involves multi-step and complex electron and proton transfer processes. It is a great challenge in this field to fully analyze the thermodynamic and kinetic factors of each catalytic process, which is rarely reported. In the recently reported CO₂-to-CH₄ conversion systems, the theoretical calculation research mainly focuses on the thermodynamic factors of CO₂-to-CH₄ intermediates (*Nat. Commun.* **2023**, *14*, 6168.), or the kinetic and thermodynamic analysis of CO₂ to *COOH and *CHO (*Nat. Commun.* **2024**, *15*, 1109). These works also identified that the formation of *COOH from *CO₂ is a rate-determining step for CO₂-to-CH₄ conversion.

At present, we are challenging the thermodynamic and kinetic simulation of the whole process of CO₂ to CH₄ conversion catalyzed by [Os] complexes, and analyzing the transition state and protonation process of each step with our cooperators, hoping to carry out theoretical calculation investigation systematically.

9. The text contains some grammar errors. The author should be more careful.

Response: Thanks for your useful advice.

We have thoroughly checked the grammar and spelling of the article again and made corresponding corrections.

Reviewer #3 (Remarks to the Author):

The author proposed an interesting photocathode for solar-driven CO₂-to-CH₄ conversion by coupling [Os] complexes with a widely available silicon. The photocathode demonstrated a considerable photocurrent density of -14.11 mA cm⁻² at 0.0 V_{RHE} with a high CH₄ selectivity of >90%. The reaction mechanism was theoretically and spectroscopically studied. However, considering the top level of Nature Communications, this work can't be accepted before the following issues are addressed. Please see below the detailed comments.

1. The underlying cause of the correlation of different catalytic rate constants between przpOs and trzpOs with their performance variation needs to be elaborated with evidence.

Response: Thanks a lot for your useful advice.

Electrochemical experiments show that przpOs and trzpOs catalysts have similar catalytic rate constants (k_{cat}) of 0.104 and 0.122 s⁻¹ without proton sources, but significantly different $k_{\text{cat-H}}$ of 0.545 and 6.41 s⁻¹, respectively, with a supplementary proton source. We hypothesized that an extra nitrogen atom on the 3,3'-bi(1,2,4-triazole) ligand in trzpOs serves as a proton-relay that facilitates the proton transfer for the sequential CO₂ reduction process. The N atom can provide binding sites for protons, which was verified in subsequent comparative experiments. Under N₂ atmosphere, as TFA was incrementally added to the acetonitrile solution of przpOs, its E₂ peak became irreversible with a notable increase in the cathodic current, pointing towards a characteristic of the hydrogen evolution process (HER) (**Figure S5**). Conversely, the HER current of trzpOs demonstrated a considerably greater enhancement compared to that of przpOs, thus confirming the superior proton-binding capability and showcasing trzpOs' heightened proton-reduction activity.

Figure S5. Electrochemical HER of [Os] complex. CV curves of 1.0 mM of przpOs (a) and trzpOs (b) in N₂ atmosphere with the addition of proton source of TFA.

Moreover, potential-dependent UV–vis spectra of przpOs and trzpOs show the formation of the reduced states of [Os] complexes and the subsequent reaction with CO₂. The thermodynamic differences between trzpOs and przpOs in the process of catalyzing CO₂-to-CH₄ conversion were compared by DFT calculations. For the rate-determining step of *CO₂ to *COOH, trzpOs has lower thermodynamic requirements, aligning with its higher catalytic activity.

In the revised manuscript, we strengthened the description of the correlation between different catalytic rate constants of przpOs and trzpOs and their performance.

2. The authors declaimed that the successful preparation of przpOs and trzpOs were confirmed by various techniques. It is suggested to elaborate the relative discussions.

Response: Thanks a lot for your useful advice.

The successful preparation of przpOs and trzpOs was confirmed by hydrogen nuclear magnetic resonance (¹H-NMR) spectroscopy (**Figure S1-S2**), mass spectroscopy (MS) (**Figure S3**) and Fourier transform infrared (FTIR) spectroscopy (**Figure S4**). The ¹H NMR spectroscopic analysis confirmed the presence of 32 protons in complexes przpOs and 30 protons in trzpOs, each associated with their respective ligands. Specifically, the ¹H NMR spectrum of przpOs displayed a distinctive singlet at 6.76 ppm, attributed to the two protons on the 3,3'-bipyrazole ligand, while the methyl protons corresponding to the PhPMe₂ ligands were observed at 0.76 (s, 12H). High-

resolution mass spectrometry (HR-MS) studies revealed a molecular mass of 916.1683 for przpOs, compared to that of 918.1588 observed for trzpOs. The molecular ion peak signals of both complexes were prominent in the HR-MS analysis (**Figure S3**), with minimal ion residue peaks suggesting their stability and resistance to decomposition into ion fragments. The stretching vibration signals of aromatic C-H and methyl C-H were confirmed in FTIR spectra, and the stretching vibration characteristics of the aromatic ring skeleton were observed (**Figure S4**). The solid-state of przpOs was further confirmed by the single-crystal X-ray structural characterization (**Figure 1b**). For the side view of przpOs, two P ligands are in the axial direction with Os-P lengths of 2.3388 and 2.3429 Å, respectively, and P-Os-P dihedral angle is 174.59°. Moreover, the phenyl group in the P ligand is parallel to the phen ligand due to the π - π stacking. From the top view, the Os atom locates at the center of the plane constructed by phen and bipyrzole ligands, and the distances between Os and N atoms are 2.0852, 2.0791, 2.0760 and 2.0649 Å, respectively (**Figure 1b**).

3. It is suggested to discuss the correlation of UV-Vis absorption of both przpOs and trzpOs with the activity in Page 6.

Response: Thanks a lot for your useful advice.

“Consequently, the optical band gaps ($E_{g,op}$) of przpOs and trzpOs calculated from the edge of MLCT absorption peaks are 2.30 and 2.45 eV, respectively (**Table S1**). Interestingly, trzpOs has a larger $E_{g,op}$ value than that of przpOs, indicating that its excited state possesses higher energy for transferring photogenerated charge to the CO₂ substrate, potentially leading to more efficient CO₂ reduction activity. This finding aligns with the results obtained from the electrochemical studies.”

4. In page 7, the catalytic current enhancement of trzpOs may be due to a broad range of reasons. It is not scientific enough to attribute the enhancement merely to that the N atoms on the triazole ligand provide binding site for protons to promote the PECT process.

Response: Thanks a lot for your useful advice.

Compared with 3,3'-bipyrazole, the additional N atom of 3,3'-bi(1,2,4-triazole) ligand can indeed be used as a reactive site and to form a significantly different catalytic CO₂ reduction mechanism. However, the results of electrochemical experiments show that the catalytic activity TOF of przpOs and trzpOs under neutral conditions is very close (0.104 vs. 0.122 s⁻¹), so it is speculated that the catalytic mechanism of the two catalysts is similar. To systematically compare the two catalysts, we use the N atom at the same position as the catalytic site.

Moreover, DFT calculations of the frontier orbital distribution also indicate that the HOMO orbitals of either the S₀ or T₁ states of przpOs and trzpOs are distributed on the N₄ atom of 3,3'-bipyrazole and 3,3'-bi(1,2,4-triazole) ligands, respectively (Figure R?). These results suggest that the binding sites of CO₂ to przpOs or trzpOs tend to be the same N₄ atom, rather than the additional N atom on the 3,3'-bi(1,2,4-triazole) ligand. Besides, the electrostatic potential distributions indicate that there are negative charge distributions on the same N₄ atom of przpOs and trzpOs, which is conducive to the nucleophilic attack reaction with CO₂ substrate.

Figure S12. The orbital distribution and electrostatic potential of singlet and triplet states. The distributions of HOMO and LUMO of przpOs in S₀ (a) and T₁ (b)

states, and trzpOs in S₀ (c) and T₁ (d) states. Electrostatic potential of przpOs in S₀ (e) and T₁ (f) states, and trzpOs in S₀ (g) and T₁ (g) states, respectively.

5. There is lack of evidence for CO₂ adduct, which may be a key intermediate of CO₂-to-CH₄ conversion.

Response: Thanks a lot for your useful advice.

The CO₂-to-CH₄ conversion is one of the most difficult conversion processes in the CO₂R reaction, which requires at least 8 electrons and 8 protons to participate in the whole reaction. The intermediate species experienced are numerous and there are complex mutual conversion networks. There are many intermediate species and complex mutual conversion networks, so it is extremely difficult to realize the detection of CO₂ adduct intermediates. At present, in-situ attenuated total reflection-surface-enhanced infrared absorption spectroscopy (ATR-SEIRAS) was usually performed to monitor the reaction intermediates in the electrocatalytic or photocatalytic system with high currents or relatively slow kinetics (*J. Energy Chem.* **2024**, *88*, 543; *Nat. Commun.* **2024**, *15*, 1109.). For the photoelectrochemical system with a low current and a small amount of catalyst, it is still a great challenge to detect CO₂ intermediates by in-situ ATR-SEIRAS.

In this work, we tried to use ATR-SEIRAS to monitor the intermediate signals of the [Os] complex catalyst on the Si-based electrode. However, the experimental results failed to detect the characteristic signal of the CO₂ adduct (Figure R3)! The following factors were analyzed: First, the photocurrent density of Si-base photocathode in this work is too low to be compared with the electrocatalytic system with a high current above 100 mA or even ampere-level current. Second, the loading amount of [Os] catalyst on the electrode surface was only 1.6 nmol/cm². (EDX analysis showed that the content of Os on the electrode surface was only 0.95 ~ 1.01 %). Third, the [Os] catalyst has a relatively larger catalytic rate constant than the traditional CO₂R catalyst. In summary, these factors lead to a very low concentration of intermediates formed by the [Os] catalyst, which is below the detection line of the current ATR-SEIRAS technology.

Figure R3. The ATR-SEIRAS spectra of Si/TiO₂/przOs photocathode were recorded in CO₂ saturated 0.1 M KHCO₃ solution under illumination. Background spectrum was taken at 0.6 V_{RHE}. Potential scan rate = 5 mV/s.

Moreover, in-situ UV-vis spectro-electrochemistry of 1.0 mM przOs and trzpOs solution in N₂ and CO₂ saturated electrolyte were performed at different applied potentials (**Figure 2f-g**). In the potential-dependent UV-vis spectra, przOs and trzpOs catalysts at high applied potentials exhibit new absorptions at 372 and 294 nm, 360 and 290 nm, respectively, in N₂ atmosphere, which is attributed to the formation of reduced state species. Furthermore, these signals decrease sharply after purging CO₂ with new peaks gradually increased, indicating the interaction between the reduced state of [Os] complex and CO₂ to form CO₂ adducts. Referring to recent reports on the conversion of CO₂ to CH₄ (*Nat. Commun.* **2023**, *14*, 6168; *Nat. Commun.* **2024**, *15*, 1109), we calculated the thermodynamics changes of common intermediates in the CO₂ reduction process to compare the activity of the two catalysts.

6. The writing needs polishment.

Response: Thanks a lot for your suggestion.

We have thoroughly checked the grammar and spelling of the article again and made corresponding corrections.

REVIEWER COMMENTS

Reviewer #1 (Remarks to the Author):

The responses were suitable, leading to an improvement in the paper's quality.

Reviewer #2 (Remarks to the Author):

The authors provided more solid evidence in the response to understand the mechanism. However, there are still two parts are not clear:

1.The transition of excited state (przpOs T1 and trzpOs T1) during the reaction process should be revealed more clearly. As the author said, the excited state will be restored to the ground state. I think the readers are curious about when this state transition happened, in which elementary step?

2.I fully understand the difficulty in the analysis of kinetic process of whole reaction. Nonetheless, the kinetic barrier of the crucial steps (e.g. CO₂ to COOH) is better to be exhibited.

Reviewer #3 (Remarks to the Author):

I am happy with the revision and would like to recommend the acceptance.

Reviewer #2 (Remarks to the Author):

The authors provided more solid evidence in the response to understand the mechanism. However, there are still two parts are not clear:

1. The transition of excited state (przpOs T1 and trzpOs T1) during the reaction process should be revealed more clearly. As the author said, the excited state will be restored to the ground state. I think the readers are curious about when this state transition happened, in which elementary step?

Response: Thanks a lot for your useful advice. The ground state and the excited state are different quantum states of the same substance. Excited states are quantum states generated under light irradiation, which have higher energy than the ground state and can be divided into singlet (S) and triplet (T) states. In photochemical theory, photochemical reactions usually occur in the S_1 and T_1 states. In this work, [Os] complexes have long lifetimes of photoluminescence at microsecond level, so its excited state is designated as the T_1 state. (S_0 generally has fluorescence emission with short lifetime at nanosecond level.) When the excited state reacts with the substrate (or electron) to produce free radicals or ions, it's neither excited state nor ground state (Figure R1). After the releasing of final product, the [Os] catalyst completes the catalytic cycle and restores to its ground state. To clearly show the intermediates and the conversion process, we modified Figure 5 and SI.

Figure R1. The conversion of the catalyst state in the catalytic cycle.

2.I fully understand the difficulty in the analysis of kinetic process of whole reaction. Nonetheless, the kinetic barrier of the crucial steps (e.g. CO₂ to COOH) is better to be exhibited.

Response: Thanks a lot for your useful advice. The kinetic protonation process of adsorbed *CO₂ *via* directly protonation (Pathway I) and *via* nitrogen heteroatom ferrying (Pathway II) were calculated.

In the revised manuscript, we made the following changes:

The conversion pathway of CO₂ initial from the ground state (S₀) or triplet excited state (T₁) of przpOs and trzpOs photocatalysts was further studied by DFT calculations.¹⁸ As shown in **Figure S13-S20**, the optimized adsorption configurations of [Os] complexes are shown for each intermediate, such as *CO₂, *COOH, *CO, *CHO, *CH₂O, *CH₃, and *CH₄, with C atoms binding to the electron deficient N atoms on the bipyrazole and triazole ligands of przpOs and trzpOs, respectively. Notably, when CO₂ adsorption occurs in the ground state of przpOs and trzpOs, the Gibbs free energy changes (ΔG) are +0.20 and +0.08 eV, respectively. Subsequently, converting the adsorbed *CO₂ in ground state (S₀) to *COOH is an endothermic reaction that must overcome an energy barrier of +2.03 and 1.91 eV for przpOs and trzpOs catalysts, respectively (**Figure 5c**). Oppositely, when [Os] complex is in T₁ state, according to its long luminescence lifetime as shown in **Figure 1d**, the Gibbs free energy for CO₂ adsorption decreases to -0.07 and -0.04 eV for przpOs and trzpOs, respectively. Correspondingly, the required energy for the conversion of *CO₂ to *COOH is dramatically decreased to +0.34 and +0.25 eV, respectively. These results suggest that the formation of *COOH intermediates is the rate-limiting step for further hydrogenation processes⁶⁰, and [Os] complexes tend to interact with CO₂ substrates through excited states and drive the subsequent conversion process.

Figure 5. DFT calculations of the CO₂ conversion pathway and the transition states of *CO₂ protonation. (a) Calculated adsorption configurations of CO₂ and reactive intermediates on przpOs (a) and trzpOs (b). (a) Gibbs free energy diagrams for CO₂ adsorption and CO₂ reduction to CH₄ by przpOs and trzpOs catalysts. Transition states of przpOs (b) and trzpOs (c) *via* directly protonation (Pathway I) and *via* nitrogen heteroatom ferrying (Pathway II).

The subsequent CO₂ hydrogenations, for instance, the kinetic protonation process of adsorbed *CO₂, are the crucial steps of the CO₂R reaction. Since the CO₂R reaction was carried out in aqueous, hydrated proton is used as the proton source for the converting *CO₂ to *COOH. As shown in **Figure 5b**, przpOs adopts a directly protonation of *CO₂ (Pathway I) with a ΔG of +0.44 eV. In contrast, trzpOs can protonate the N atom on triazole ligand first and then transfer protons to *CO₂,

which is a two-step protonation *via* nitrogen heteroatom ferrying (Pathway II). The transition state energy barrier of two-step protonation of trzpOs is only +0.40 and +0.04 eV, which is more favorable than the direct protonation process (**Figure 5c**). These results confirm that the extra nitrogen atom on triazole ligand could serve as a proton-relay to facilitate the proton transfer for the sequential CO₂ hydrogenations. Afterwards, the intermediate exhibits a downhill free energy change at present of protons and electrons, e.g., the transformation of *CO to *CH₄ and the release of CH₄ are thermodynamically exothermic reactions which can be spontaneously carried out by receiving photogenerated electrons from the Si photocathode (**Figure 5a**). DFT results indicate that CH₄ is the most easily desorbed product from the [Os] complexes, resulting in the high selectivity for CO₂-to-CH₄ conversion. In contrast, the required energy for the hydrogenation process from *COOH to *CH₃ of trzpOs is smaller than that of przpOs.

REVIEWERS' COMMENTS

Reviewer #2 (Remarks to the Author):

Authors have well addressed my comments in last review. I think it can be accepted for publication from my point of view.

Reviewer #2 (Remarks to the Author):

Authors have well addressed my comments in the last review. I think it can be accepted for publication from my point of view.

Response: Thanks a lot for your useful advice.